# Advancing Neuroscience and Therapy: Insights into Genetic and Non-Genetic Neuromodulation Approaches

**DOI:** 10.3390/cells14020122

**Published:** 2025-01-15

**Authors:** Weijia Zhi, Ying Li, Lifeng Wang, Xiangjun Hu

**Affiliations:** 1Beijing Institute of Radiation Medicine, Beijing 100850, China; zhiweijia@bmi.ac.cn; 2School of Life Science and Technology, Xi’an Jiaotong University, Xi’an 710049, China; liying_2024@stu.xjtu.edu.cn

**Keywords:** genetic neuromodulation, magnetogenetics, sonogenetics, non-genetic physical neuromodulation

## Abstract

Neuromodulation stands as a cutting-edge approach in the fields of neuroscience and therapeutic intervention typically involving the regulation of neural activity through physical and chemical stimuli. The purpose of this review is to provide an overview and evaluation of different neuromodulation techniques, anticipating a clearer understanding of the future developmental trajectories and the challenges faced within the domain of neuromodulation that can be achieved. This review categorizes neuromodulation techniques into genetic neuromodulation methods (including optogenetics, chemogenetics, sonogenetics, and magnetogenetics) and non-genetic neuromodulation methods (including deep brain stimulation, transcranial magnetic stimulation, transcranial direct current stimulation, transcranial ultrasound stimulation, photobiomodulation therapy, infrared neuromodulation, electromagnetic stimulation, sensory stimulation therapy, and multi-physical-factor stimulation techniques). By systematically evaluating the principles, mechanisms, advantages, limitations, and efficacy in modulating neuronal activity and the potential applications in interventions of neurological disorders of these neuromodulation techniques, a comprehensive picture is gradually emerging regarding the advantages and challenges of neuromodulation techniques, their developmental trajectory, and their potential clinical applications. This review highlights significant advancements in applying these techniques to treat neurological and psychiatric disorders. Genetic methods, such as sonogenetics and magnetogenetics, have demonstrated high specificity and temporal precision in targeting neuronal populations, while non-genetic methods, such as transcranial magnetic stimulation and photobiomodulation therapy, offer noninvasive and versatile clinical intervention options. The transformative potential of these neuromodulation techniques in neuroscience research and clinical practice is underscored, emphasizing the need for integration and innovation in technologies, the optimization of delivery methods, the improvement of mediums, and the evaluation of toxicity to fully harness their therapeutic potential.

## 1. Introduction

Since the turn of the century, the field of neuroscience has undergone a remarkable transformation, fueled by a surge in technological innovations. Among these advancements, optogenetics has emerged as a cornerstone, revolutionizing our ability to manipulate neural activity with unprecedented precision [1]. The advent of optogenetic technology has paved the way for a diverse array of neuromodulation techniques, including sonogenetics, magnetogenetics, and more [2]. These genetics-based approaches have not only propelled basic neuroscience research to new heights but have also shown immense promise in clinical applications.

On the other hand, given the severely high prevalence and disability rate of mental diseases in all countries, drug treatment can no longer meet existing needs, and developing treatment methods based on the exact pathogenesis of diseases is urgent [3]. In response, researchers have turned to neuromodulation techniques that harness physical stimuli, offering noninvasive and genetically non-modifying alternatives. These non-genetic neuromodulation methods have garnered considerable attention in both research and clinical settings, presenting a complementary avenue for advancing our understanding of brain function and addressing therapeutic challenges.

The convergence of genetic and non-genetic neuromodulation approaches represents a pivotal moment in therapeutics, as it integrates diverse methodologies to manipulate brain activity with precision and specificity. This review aims to elucidate the synergistic relationship between these methodologies, exploring their respective stimulus factors and regulatory mechanisms. By examining the current landscape of neuromodulation research, we seek to identify emerging trends and potential avenues for the future development of optimized genetic neuromodulation techniques. Through a comprehensive analysis and synthesis, this review endeavors to provide valuable insights into the evolving field of neuromodulation. By bridging the gap between fundamental research and clinical translation, we aspire to catalyze transformative advancements in both neuroscience and medicine, ultimately improving outcomes for individuals affected by neurological disorders.

## 2. Genetic Neuromodulation: Tools and Strategies

Remote, rapid, and reversible regulation of cell activity provides convenient conditions for studying and elucidating unknown physiological processes. In the past decade, physical and chemical genetics technologies represented by optogenetics have been increasing and have been widely used in the regulation of cell activity. Taking the application of optogenetics, a technique that uses light to control cells in living tissue, typically neurons, as an example, since 2010, the number of related publications has shown exponential growth [4]. This kind of neuromodulation technology combines genetics and physical/chemical methods, and its construction and realization can be summarized into three elements: (1) physical/chemical factor stimulation; (2) medium for sensing physical/chemical factors; and (3) method of delivering the physical/chemical stimulus to target cells/areas. Specifically, first, the corresponding sensory proteins are targeted and expressed in specific neurons using virus transfection and similar technologies, enabling them to convert physical signals into electrical currents or trigger cascade reactions through signal transduction. Then, physical/chemical stimuli are used to intervene in the subject when a specific physiological/pathological process occurs to induce or inhibit its biological process or behavior. Currently known proteins capable of sensing physical stimuli include photoreceptor proteins, mechanosensitive proteins, and the transient receptor potential (TRP) protein family, which consists of ion channels located mostly on the plasma membrane of numerous animal cell types and can be activated by temperature. Through a deep dive into protein structure and function and continuous advancements in virus genetic engineering, neuromodulation techniques using light, ultrasound, and magnetothermal stimulation have advanced significantly (see Figure 1). However, it is crucial to recognize that these methods come with their own pros and cons, including differences in precision, potential for tissue trauma, and specificity in targeting. These distinctions heavily influence their popularity and future applications.

### 2.1. Optogenetics: Precision Control with Light

Optogenetics entails the introduction of light-sensitive genes (such as *channelrhodopsin-2* (*ChR2*), engineered bacteriorhodopsin (eBR), *natronomonas halorhodopsin 3.0* (*NpHR3.0*), *archaeal halorhodopsin* (*Arch*), or *optogenetic G-protein-coupled receptor* (*OptoXR*)) into specific neurons, enabling precise control of their activity with light. These genes encode proteins that can be activated or inhibited by specific wavelengths of light. For instance, ChR2 is a channelrhodopsin that, when exposed to blue light, opens ion channels and depolarizes the neuron, leading to action potentials. Conversely, NpHR3.0 is a halorhodopsin that can hyperpolarize neurons when illuminated with yellow light, effectively silencing their activity. By using these light-sensitive proteins, researchers can precisely control the timing and activity of targeted neurons in living tissues, allowing for detailed studies of neural circuits and behaviors.

Recently, optogenetics has been widely used in genome editing [5], the activation and inhibition of genes (e.g., preventing oncogene-induced tumor formation) [6], synaptic communication (e.g., using the gene-encoded gap junction probe to detect the communication characteristics between specific cells in cardiomyocytes) [7], the regulation of the activity of specific neurons, and the exploration of the function of specific neural circuits under physiological and pathological conditions in the study of different functions and their potential regulatory mechanisms. On a spatial scale, optogenetics works at the subcellular level (e.g., using light to manipulate microtubule positioning and translocation from the level of cytoskeletal dynamics) [8], at the cellular level (e.g., using biological tools to induce necrosis and pyroptosis) [9], at the level of microcircuits in single brain regions (e.g., using optogenetics to validate the important role of the inhibitory/disinhibitory microcircuits consisting of parvalbumin-positive and somatostatin-positive neurons in the amygdala in associative learning) [10], at the level of neural circuits across brain regions (e.g., the activation of neural circuits from the ventral hippocampus to the anterior limbic cortex by promoting fear extinction) [11], and at the level of global brain regions (e.g., the activation of the striatum to elicit body-rotating behavior in mice) [12]. In summary, optogenetics provides a new research tool for gaining insights into cell signaling networks and exploring the underlying neurobiological mechanisms of brain diseases.

Optogenetics allows us to control specific cells in the body incredibly quickly, but there is a catch: light does not travel well through tissues on its own. To get around this, researchers use thin, flexible fibers to guide light to deeper areas within the body. Unfortunately, this means the areas we can control are restricted to those near the fibers. While this gives us precise control over specific locations, it is not practical for influencing cells that are spread out. Additionally, inserting these fibers into the body is an invasive procedure. It can potentially cause lasting harm to tissues and trigger ongoing immune responses. One particular challenge is the impact on animal behavior studies, especially those that focus on social interactions. When an animal is connected to a fiber, it is like being on a leash—it restricts its natural movements and can interfere with how it interacts with others. This is a significant concern because it can alter the very behaviors we are trying to study. Furthermore, constantly shining light on tissues can cause damage over time, as well as unintended effects from heat generated by the light and the desensitization of the light-sensitive proteins we use to see the cells. This desensitization can make the proteins less effective or even stop them from glowing altogether.

Given these concerns, optimized optogenetic technology based on wireless optogenetics was developed. The rapid development of the nanomanufacturing industry, mechanical engineering industry, and material engineering industry has provided good technical support for the development of wireless optogenetics platforms. The iteration of the existing technology involves the improvement of photosensitive proteins (e.g., the application of redshifted opsins) [13], the application of nanoparticles (e.g., optogenetic antennas based on upconverting micro/nanoparticles) [14], the development of a closed-loop optofluidic system (or closed-loop optogenetics) [15], the fusion of acousto-optogenetic techniques using mechanosoluble nanoparticles [16], and the development of non-genetic optical neural interfaces [17]. Redshifted opsin is an opsin that can be activated by red light. The discovery of redshifted opsin has greatly expanded the existing opsin tool library. Compared with blue-light-activated channelrhodopsin, redshifted opsin can be activated at a longer wavelength, which allows researchers to combine redshifted opsin, channelrhodopsin, and genetically encoded Ca^2+^ indicators (GECIs) to explore and manipulate complex neural mechanisms simultaneously [18]. To date, a variety of redshifted opsins are available, such as the redshifted opsin volvox channelrhodopsin-1 (VChR1) in volvox [19] and Chrimson with longer redshifted properties [20].

Nanoparticles are materials with at least one dimension measuring in the billionths of a meter (nanometers). These tiny scales give them special characteristics, such as a large surface area relative to their volume, which enhances their reactivity and allows them to interact with biological systems in ways that larger particles cannot [21,22]. An optogenetic antenna based on the upconversion of micro/nanoparticles, a process where these particles absorb multiple low-energy photons (such as near-infrared light) and emit a single higher-energy photon (such as visible or ultraviolet light), overcomes the shortcomings of traditional optogenetics technology [14]. In this photophysical process, micro/nanoparticles doped with rare-earth ions sequentially absorb two or more photons of lower energy, collectively exciting the electrons to higher energy states [23]. When these electrons relax back to their ground state, they release the accumulated energy as a single higher-energy photon [24]. This mechanism allows for deeper tissue penetration with minimal damage, enabling wireless and more precise neuromodulation.

In the past, most optogenetic neuromodulation technologies used open-loop optogenetic systems to directly activate or inhibit the activity of targeted neurons and observe the associated behavioral changes, while closed-loop optogenetic stimulation devices provided an efficient means to study the causal relationship between neural circuits and related phenotypes [25]. Closed-loop optogenetics exploits the synchronicity of neural activity or behavior to guide stimulation by sensing indicators in a closed feedback loop to make real-time decisions about how and when to deliver optogenetic stimulation. For example, Xu et al. (2022) used closed-loop optogenetics to screen the corresponding brain regions involved in licking. In this experiment, closed-loop optogenetics inhibited the activity of neurons in different brain regions with the same probability at different times [26]. To achieve noninvasive deep brain optogenetic stimulation, the researchers also used mechanoluminescent materials generated by focused ultrasound to activate opsin-expressing neurons. This technique is called acousto-optogenetics (or sono-optogenetics) and is based on mechanically soluble nanoparticles. Recently, Wang et al. developed a liposomal (Lipo@IR780/L012) nanoparticle that can circulate in the blood through intravenous injection and cause a cascade reaction under the stimulation of focused ultrasound. In turn, synchronous and stable blue light is generated and acts on opsin-targeting motor cortex neurons, allowing for the manipulation of mouse limb movements [16]. In addition, due to the rapid development of semiconductor micro/nanoprocessing technology, some studies have also investigated using the physical and chemical properties of flexible silicon products (for example, a silicon mesh that can be attached to the upper back of the cerebral cortex) to achieve neuromodulation. Relevant studies have systematically explained the feasibility of silicon-based biointerface materials in non-genetic light-controlled neural modulation of cell and neuron activity and laid a solid foundation for the realization and optimization of brain–computer interfaces [17].

A recent study introduced a wearable miniature optogenetic pacemaker that enables real-time, highly accurate monitoring of the heartbeats of freely moving mice and can specifically control the pumping of ChR2 to the heart. This device uses blue light to activate ChR2, a light-sensitive protein, which then precisely regulates the contractions of the heart muscle, allowing for direct and noninvasive control of the heart’s pumping action. These methods facilitate simultaneous studies of the body and the brain, exploring the central (e.g., insular cortex) and peripheral (e.g., cardiac interoceptive) neural mechanisms. Researchers have also developed BiPOLES, an optogenetic tool for different types of neurons that can simultaneously achieve two-color and two-directional regulation of the same position. With this tool, blue light stimulation at 460 nm can elicit excitatory postsynaptic currents in interneurons but has no effect on the currents of vasoactive intestinal peptide (VIP)-expressing neurons, whereas orange light stimulation at 635 nm elicits postsynaptic inhibitory currents in VIP neurons but has no effect on interneurons [27]. The study also found that the use of different wavelengths of light to activate or inhibit glutamatergic neurons expressing somBiPOLES (an optimized BiPOLES) in *Drosophila melanogaster* larvae causes relaxation or contraction of the body, respectively; light activation or inhibition of dopamine neurons expressing somBiPOLES in the mouse locus coeruleus causes a reduction in or dilation of the pupil diameter, respectively [27]. Excitingly, optogenetics is also expected to be applied to clinical treatment. For instance, optogenetics has shown promise in the treatment of neurodegenerative diseases [28]. Fernández-García et al. (2020) demonstrated the application of optogenetics in reversing motor symptoms and synaptic deficits in Huntington’s disease (HD). In their study, they used optogenetic stimulation targeting the M2 cortex–dorsolateral striatum circuit in a mouse model of HD. The parameters included stimulation at 20 Hz, with a 5 ms pulse width, and a total duration of 30 min per day over a period of 14 days. This treatment resulted in significant improvements in motor function, as well as the restoration of synaptic plasticity and neurotransmitter balance in the affected brain regions. These findings underscore the potential of optogenetic approaches in modulating specific neural circuits to alleviate symptoms and restore normal function in neurodegenerative diseases [28,29]. Additionally, optogenetics technology has been successfully applied to restore the visual function of patients with retinitis pigmentosa [30,31]. The goal of curing diseases through a beam of light will no longer be a dream.

In short, optogenetics has achieved significant milestones, and ongoing research aimed at addressing its limitations continues to emerge. This technology has evolved into a vital and indispensable tool for elucidating cellular and subcellular functions, as well as the diverse mechanisms of information transfer within tissues and organisms. Noninvasive optogenetic methods ensure the investigation of internal mechanisms within living tissues without causing additional harm to the body. Micro/nanoscale optogenetic devices offer valuable references and convenience for potential clinical applications. Closed-loop optogenetics introduces new possibilities for orchestrating intricate regulatory processes in vivo, and promising results from clinical trials hint at the potential clinical translation of optogenetics. However, several areas require continuous refinement in the future (as outlined in Table 1). This includes optimizing high-throughput computational quantification techniques for in vivo optogenetic data and clarifying precise experimental parameters. Additionally, enhancing the performance of light-sensitive tools to achieve more precise targeting of deep brain regions or nuclei is necessary. Expanding optogenetic applications beyond the brain, such as in the heart or bone, and exploring the interplay between these body tissues and the brain in tandem with studies on brain function remain critical avenues for further investigation.

### 2.2. Chemogenetics: Commanding Neurons with Compounds

Chemogenetics, also known as pharmacogenetics, is a technique that involves the modification of specific biological macromolecules, such as receptors or proteins, to make them responsive to otherwise nonbinding biological small molecules. This allows for the precise control of cellular activities using specific ligands or substrates. Chemogenetics emerged in the 1990s as an important neurobiological research tool comparable to optogenetics. In recent years, the synergistic integration of optogenetics and chemogenetics has emerged as a formidable tool for elucidating the neural mechanisms behind precise cognitive behaviors, such as memory processes, or understanding abnormal neuropsychiatric disease manifestations, like fear-induced trembling. This combined approach also holds immense promise in unraveling intricate cellular signaling pathways, facilitating drug development, and establishing robust platforms for functional genomics studies. Chemogenetics works by modifying certain biological macromolecules to interact with otherwise nonbinding biological small molecules [32]. These recombinant proteins or channels can be activated by highly specific ligands or substrates to achieve reversible, controllable modulation of neural activity. Although chemogenetics lacks the advantage of high spatial and temporal resolution seen in optogenetics, the closer level of control achieved by chemogenetics is a better fit with the need to probe the long-term regulation of neural circuits. In addition, numerous Food and Drug Administration (FDA)-approved drugs target G-protein-coupled receptors (GPCRs) [33], and designer receptors exclusively activated by designer drugs (DREADDs), which are tools for chemogenetics, are modified GPCRs. Thus, this technology has greater potential for clinical applications.

Chemogenetics can regulate cellular activities through two primary signaling modes: ionic signaling, which involves ion-channel-mediated changes in membrane potential, and metabotropic signaling, which engages GPCRs to activate intracellular signaling cascades. Metabotropic signaling utilizes engineered receptors to activate effector proteins and initiate multiple downstream signaling cascades. In 1991, Strader et al. first constructed the allele-specific activation of genetically engineered receptors, which was the first GPCR-based chemical genetics tool [34]. To date, a variety of successful biomolecular receptors have been created, including, but not limited to, receptors activated solely by synthetic ligands (RASSLs), genetically engineered receptors, DREADDs, and many other biomolecular receptors that have been successfully modified. The most widely used receptors in chemogenetics are DREADDs. The method involves changing the structure of GPCRs to ensure that the receptors can only be activated or inhibited by specific compounds. The modified receptors can achieve different purposes of cell activity regulation through various GPCR cascade reactions (Gq, Gi, Gs, and β-arrestin).

Generally, current research uses Gq-DREADDs and Gi-DREADDs to interact with the compound clozapine-N-oxide (CNO) through local intraperitoneal injection or oral administration to activate or inhibit the activity of cells or neurons [35]. Gq-DREADDs, also known as human muscarinic acetylcholine receptor subtype M3 (hM3Dq), are modified from human muscarinic acetylcholine receptor subtype M3 (hM3) [36]. Under typical physiological conditions, GPCRs, such as hM3, exhibit the ability to recognize a wide array of ligands, such as acetylcholine. Through canonical conformational changes, they facilitate interactions with various heterotrimeric G protein families, including Gq, Gi, and Gs, as well as signaling molecules independent of G proteins. This orchestration allows them to regulate intracellular responses, adapting to fluctuating environmental conditions. However, when the Y149C^3.33^/A239G^5.46^ sites of hM3 were mutated, hM3 no longer bound acetylcholine but reacted with nanomolar concentrations of CNO [37,38]. This mutated hM3 receptor is named *hM3Dq* and mainly plays an activating role. Similarly, human muscarinic acetylcholine receptor subtypes M2 (hM2) and M4 (hM4) can activate the downstream Gi pathway, and CNO dose-dependent effects were also observed after mutations of the Y149C^3.33^/A239G^5.46^ sites. The mutated hM2 and hM4 receptors, named hM2Di and *hM4Di*, mainly play an inhibitory role. After site-directed mutagenesis, Gs-DREADDs (or rM3Ds) regulate neuronal activity by coupling to gas signaling [39]. Moreover, alongside chemogenetic modulation achieved through the viral injection of adeno-associated viruses (AAVs; a small virus that infects humans and some other primate species, often used as a vector for gene therapy due to its ability to transduce non-dividing cells and its low immunogenicity) or lentiviruses expressing specific subtypes of DREADDs targeted to particular brain regions or cell populations, precise control over the target cell populations is now attainable through the creation of transgenic mice that carry Cre-dependent *hM3Dq* and *hM4Di* constructs [40].

Currently, various in vivo studies have reported diverse effects induced by DREADDs. For example, activation of Gq-DREADD-based acute chemogenetics (onset 2 h after 1 mg/kg CNO administration) significantly inhibited food intake levels in glucose-dependent insulinotropic polypeptide receptor (Gipr)-Cre mice, illustrating the significance of hypothalamic Gipr neurons in anorectic behavior [41]. Furthermore, a study revealed that the introduction of *hM4Di* into the ventral prefrontal cortex led to a significant reduction in depressive behavior among Pvalb-Cre mice [42]. Electrophysiological findings further indicated that this inhibitory effect was mediated by two specific GABAergic interneurons, namely, somatostatin and parvalbumin [42]. Injection of rM3D(Gs)-DREADD into the locus coeruleus can modulate pain-induced depressive symptoms [43]. In addition to using chemogenetics to regulate the behavior of rodents, chemogenetics based on the DREADD system has also been used in Drosophila [44], monkeys [45], and other organisms in studies. In addition, the technique has been shown to modulate the activity of many types of neurons. For example, astrocytes are closely related to synaptic repair functions, such as dendritic spine renewal [46]. Expression of *hM3Dq* receptors by local injection of AAVs in the somatosensory cortex (S1) increases the occurrence of Ca^2+^ transients in astrocytes and the upregulation of acidic protein levels in glial protofibrils, showing a sustained reversal of pain behavior [47]. DREADDs have also been used to control the GPCR signaling cascade response in a variety of cells, including hepatocytes and pancreatic β-cells [48].

In summary, the utilization of chemogenetics, particularly with DREADDs, has proliferated and become a cornerstone in the toolkit of methods for deciphering precise cognitive and behavioral patterns, as well as for modulating neuronal activity and signaling. Nevertheless, there are still numerous pressing challenges that demand immediate attention. For instance, a notable challenge arises from the fact that exogenous compounds (e.g., CNO) cannot directly act on ion channels but rather exert their effects by initiating downstream cascade reactions of GPCRs, rendering their effects less stable. Modified biomolecules may exhibit significant affinity for natural receptors, leading to competition with the target receptor and potentially diminishing the intended regulatory effect. Therefore, strategies like gene knockout are often necessary to avoid such nonselective activation. To address these limitations, pharmacologically selective actuator module/pharmacologically selective effector molecule (PSAM/PSEM)-based chemogenetics has emerged as a promising alternative. This approach revolves around the concept of ionotropic signaling achieved through the engineering of ligand-gated ion channels, such as the ligand-binding region of the alpha 7 nicotinic acetylcholine receptor (α7nACHR). This engineering enables direct regulation of ion conductance at the cell membrane, providing precise control over the opening or closing of ion channels and thereby regulating neuronal activity. Recently, researchers screened and optimized PSAM components from 44 clinical drugs and determined that varenicline was the PSAM component with the best selectivity, the fewest side effects, and the highest efficacy [49]. However, the role of PSAM/PSEM in nonexcitatory cells is very limited. There are also studies employing D-amino acid oxidase (DAAO)-based chemogenetics to regulate intracellular H_2_O_2_-mediated pathways [50]. H_2_O_2_ plays an important role in metabolic redox control. When its concentration is too high, it will induce oxidative stress in cells or tissues and cause serious oxidative damage to the body. Currently, DAAO has been widely used to detect the level of H_2_O_2_ in cells and subcellular structures, to regulate the amount of H_2_O_2_ in different organelles in vascular endothelial cells [51], and to construct animal models of heart failure [52]. In addition to the improved methods mentioned above, the temporal resolution of chemogenetics can be optimized from a pharmacokinetic point of view for the gradient modulation of cellular activity for different research purposes in the future.

### 2.3. Sonogenetics: Influencing Neural Activity with Sound Waves

Sonogenetics refers to the modification of genetically engineered molecules to make them sensitive to sound waves, and then the synthetic molecules are transferred into the corresponding target cells or target neurons to induce cell-type-specific regulation. Recent years have witnessed remarkable advancements in biotechnology, particularly in the application of ultrasound (with a frequency > 20 kHz), within the realms of bioengineering technology and medical diagnostics. In comparison to optogenetics, ultrasound has emerged as a promising method for non-destructive modulation of cellular activity. This distinction underpins the widespread clinical use of ultrasound for both diagnosis (utilizing low-intensity ultrasound) and treatment (employing high-intensity ultrasound to disintegrate stones), exemplifying its pivotal role in medical practice. The further integration of acoustic and genetic technology has promoted the birth of sonogenetics.

Sonogenetics is closely related to the involvement of the TRP protein family, which play a crucial role in mediating cellular responses to sound stimuli, thus allowing for precise control of neuronal activity through ultrasound. TRP ion channels were first identified in Drosophila in the late 1960s [53], and the first human homolog was reported in 1995 [54]. TRP channels are distributed in both the nervous and nonneural systems, but their expression is much lower in the central nervous system than in the peripheral nervous system and other tissues and organs. They are expressed as physical and chemical sensors on the plasma membrane of various cell types, including neurons, and they can sense stimuli such as temperature, voltage, pressure, ligands, and osmolarity [55,56]. Approximately 30 *TRP* genes and more than 100 TRP channels have been identified in different species [57]. There are seven subfamilies of TRPs, including TRPC (canonical), TRPV (vanilloid), TRPM (melastatin), TRPA (ankyrin), TRPP (polycystin), TRPML (mucolipin), and TRPN (Drosophila NOMPC), six of which are found in mammals (except TRPN). In yeast, an eighth TRP family was recently identified and named TRPY [54]. TRPs share common structural features, including six putative transmembrane structural domains and intracellular C and N termini, with the fifth and sixth transmembrane domains together forming the channel pore region and the first–fourth transmembrane domains constituting the voltage-sensing sites [58]. The resolution of its structure and function found that different structural domains of the TRP channel mediate its multimodal sensitivity [59].

Ibsen et al. first proposed the concept of sonogenetics in 2015 [60]. They found that low-pressure ultrasound (the frequency of ultrasound commonly used in medical examinations) can activate mechanosensitive neurons of *Caenorhabditis elegans* expressing the *transient receptor potential-4 (TRP-4)* gene by regulating the opening of ion channels in the cell membrane [60]. Although the regulation of mammalian neurons cannot be achieved because the action of *TRP-4* is not effective in mammals, this discovery suggests the potential for sonogenetic regulation of cellular activities. In recent years, Yang et al. (2021) further used local tissue heating caused by short pulses generated by low-intensity focused ultrasound (LIFUS), a technique that uses focused ultrasound waves at low intensities to modulate neuronal activity, to demonstrate in specific types of neurons in mice selectively expressing *transient receptor potential vanilloid1 (TRPV1)* that ultrasound can be used to activate genetically encoded ion channels, achieving deep brain stimulation for stable control of behavior in free-ranging mice [61]. Based on this, the use of acoustic waves to regulate cell activity and even the behavior of living animals has become a reality.

The mechanisms of action of using sonogenetics to induce changes in cell activity (see Table 2) mainly include the following: (1) Using mechanosensitive ion channel (MSC) proteins as media for cells to perceive ultrasonic vibrations and regulating cell activity through acoustic stimulation. Specifically, the gene expressing MSC protein is transferred into target cells by virus transfection technology, and the target cells are induced to specifically express MSC protein. Then, the mechanical force generated by sound waves is used to activate MSCs to an open state. The ion flow state inside and outside can change the membrane potential, thereby activating or inhibiting cells and completing the regulation of cell activity. (2) Relying on locally responsive ultrasound (heat)-sensitive proteins/nanoparticles to regulate cell activity. Ultrasound-sensitive proteins or nanoparticles are introduced into target cells and regulate cell activities by directly acting on ion channel components to induce transmembrane ion gradients. The following will introduce the two mechanisms of ultrasonic regulation of cell activity.

#### 2.3.1. Using Acoustic Waves to Regulate the Activation of Mechanosensitive Ion Channels in Cells

As early as 1929, Harvey proposed that a frog heart and sciatic nerve soaked in Ringer’s solution could sense ultrasonic waves. Although the latter effect was weak, this discovery provided the possibility to explore the method of regulating cell activity based on acoustic waves [68]. Before the concept of sonogenetics was proposed, ultrasound had already been used as an important stimulus in the field of neuromodulation, a technique also known as ultrasound neuromodulation. Ultrasound neuromodulation technology can use ultrasonic waves of different intensities, frequencies, pulse repetition frequencies, pulse widths, and durations to activate or inhibit neural activity at the stimulation site, thereby achieving reversible changes in neural function due to two-directional regulation. For example, the action potential of the bullfrog sciatic nerve terminal can be adjusted by changing the ultrasound parameters. Specifically, a shorter pulse duration can enhance the amplitude and rate of the action potential, and a longer pulse duration can inhibit the activity of cells. In 2008, Tyler et al. proved that low-frequency and low-voltage ultrasound induced neural activity through electrophysiological experiments on mouse brain slices and proposed a possible regulatory mechanism, that is, ultrasound affects voltage-gated sodium and calcium channels [69]. Later, the team demonstrated for the first time the possibility of using low-frequency and low-voltage ultrasound to regulate neuron activity through live animal experiments. They found that ultrasound stimulation of neurons in the motor cortex was sufficient to induce motor behavior in mice [70]. Furthermore, ultrasound neuromodulation technology has been shown to enhance the performance of sensory discrimination tasks by stimulating the human primary somatosensory cortex [71]. Excitingly, Duque et al. recently discovered the acoustic-sensitive protein TRPA1 in mammalian cells by screening one by one and further used ultrasound stimulation to achieve sonogenetic regulation of the mammalian motor cortex. They proposed that the sonosensitization of TRPA1 is related to the N-terminal domain, actin cytoskeleton, and cholesterol interaction [63]. Ultrasound can also induce a reversible torpor-like state in mice and rats by activating neurons highly expressing transient receptor potential melastatin 2 (TRPM2) in the preoptic area (POA) of the hypothalamus. Specifically, after ultrasonic stimulation, the skin temperature of the mouse scapula (the main distribution area of brown fat) decreased significantly, and the temperature of the tail (mainly responsible for heat dissipation) increased significantly [65]. Of more immediate clinical significance, ultrasound has been shown to significantly alleviate motor symptoms and improve motor coordination in mice with Parkinson’s disease (PD) [66]. This result brings greater possibilities to advance the transition of sonogenetics from modulating neuronal activity in rodents to clinical treatment of diseases related to neurological dysfunction. In short, ultrasound neuromodulation technology utilizes the opening and closing of endogenous or exogenous MSCs on cells to be stimulated under the action of mechanical force generated by acoustic waves. At the same time, the above evidence also illustrates the feasibility of ultrasound as a stimulation method in neuromodulation and clinical treatment.

Based on this, sonogenetics uses mechanosensitive proteins as media for cells to perceive ultrasonic vibrations, aiming to regulate cell activity through acoustic stimulation. The currently known mechanosensitive proteins include mechanosensitive channel of large conductance (MSCL) [64], Piezo ion channel [72], Prestin ion channel [73], TRP ion channels [65], TWIK-related K^+^ channel (TREK) [74], two-pore domain potassium channels (K2Ps), voltage-gated channels (VGCs) and MEC ion channels [75]. In contrast to traditional voltage-sensitive ion channels or ligand-gated ion channels, MSCs are able to induce changes in membrane tension by sensing cellular deformation, which, in turn, induces the opening of ion channels in the cell membrane, facilitating the transmembrane transport of ions inside and outside the cell and mediating a variety of life activities. Currently, there are two explanatory models for the biophysical principles of the gating mechanism of MSCs. The “lipid force” model posits that mechanical stimulation, specifically the tension exerted on the cell membrane by lipids in the surrounding environment, can directly impact and activate channel proteins located on the cell membrane [76]. Different types of ion channels have different activation tension ranges and are affected by the composition of the lipid bilayers. The “filamentary force“ model serves as a complementary explanation to the “lipid force” model, particularly focusing on ion channels that are not regulated by lipid force mechanisms. An example of such channels can be found in the mechanical receptor potential channels in certain insect cells. This model delves into how mechanical forces are transmitted through filament-like structures within these cells, offering an alternative perspective on how ion channels can be activated in specific contexts [77]. This model suggests that the opening of MSCs is closely related to the conduction force of the cytoskeleton, which refers to the force transmission through the cytoskeleton network that includes microtubules, actin filaments, and intermediate filaments [78]. The cytoskeleton acts as a mechanical integrator, where forces generated at the cell membrane are relayed through these filamentous structures to induce conformational changes in MSCs. This process, known as mechanotransduction, involves the dynamic interaction between the cytoskeleton and extracellular matrix (ECM) proteins, effectively converting mechanical stimuli into electrical or biochemical signals within the cell [78].

The effectiveness of sonogenetics to utilize mechanosensitive proteins to achieve noninvasive modulation of cell activity has been demonstrated in different models. For example, sonogenetics has been shown to regulate cellular activity in living animals [65], and primary hippocampal neurons expressing MscL can be activated by low pressure [64]. Focused ultrasound neuromodulation is a type of sonogenetics that has been extensively studied, using focused transducers to direct ultrasound to the target brain area or target cells. Focused ultrasound can be divided into high-intensity focused ultrasound and LIFUS. However, the thermal effect of high-intensity focused ultrasound is significant. Excessively high temperatures (exceeding the maximum temperature of 47 °C for mammals) will reduce the efficiency of enzymes and proteins and even cause denaturation or death [79]. Long-term, high-intensity sound pressure may also cause severe hearing damage and may even involve the lungs, stomach, liver, and other organs [80,81]. Therefore, it is not suitable for neuromodulation. LIFUS can exert the inherent good propagation direction, strong penetrating ability, and good focusing effect of ultrasound with high temporal (stimulus latency is less than 1 ms) and spatial accuracy (spatial resolution reaches the order of mm). The average temperature increase caused in biological tissues has also been proven to be less than 1 °C [70]. However, despite the high spatial accuracy of focused ultrasound neuromodulation, there are still many problems with this technique (Table 1). As an illustration, ultrasound possesses the ability to safely and noninvasively penetrate thin bone and tissue, with the added advantage of precise focusing within a volume as small as a few cubic millimeters. This eliminates the necessity for implantation to deliver the treatment. However, it is worth noting that it still requires the use of a fixation device in the animal’s head to ensure accurate targeting and consistent results. The acoustic pressure of the target cells is also influenced by the relative position to the transducer. Therefore, there is a pressing need to enhance the reproducibility of experiments in this field. Additionally, it is critical to strike a delicate balance in the stimulus intensity of ultrasound, considering its mechanical effect, thermal effect, and cavitation effect. Furthermore, previous studies have typically utilized ultrasound wavelengths exceeding 500 µm, a dimension substantially larger than the size of neurons, which typically measure around 20 µm. Consequently, there is a significant possibility that ultrasonic energy may not efficiently couple with neurons during the process of ultrasonic neuromodulation, leading to suboptimal regulatory outcomes. Furthermore, compared to other ion channels, the pharmacological properties of MSCs have not been fully elucidated (e.g., no blockers specifically targeting Piezo1 or agonists specifically targeting Piezo2 have been identified) [82].

#### 2.3.2. Using Ultrasound-Sensitive Proteins/Nanoparticles to Regulate the Activity of Local Responsive Cells

Sonogenetics can also rely on nanoparticles that respond locally to ultrasound to modulate cellular activity. Following the landmark cloning of TRPV1 by Professor David Julius and his research team at the University of California, San Francisco in 1997, subsequent research efforts led to the discovery of a multitude of TRP channels that are responsible for sensing diverse temperature ranges. These temperature-sensitive TRP channels are collectively known as Thermo-TRP. Thermo-TRP channels can be activated by both millisecond temperature jumps and slow temperature increases. Based on the thermosensitive properties of TRPV1, Yang et al. used ultrasound-induced thermal effects to regulate the activity of neurons [61]. Their approach involved the injection of a viral vector expressing TRPV1 into specific types of neurons within the mouse motor cortex. Subsequently, the mice received LIFUS stimulation, which induced rapid local tissue heating. This localized heating had the effect of activating the expression of TRPV1 channel neurons, allowing for the manipulation of behavior in freely moving mice. Alongside this experimentation, the researchers conducted safety assessments by detecting markers of neuronal apoptosis and employing other techniques to ensure the safety of the technology. This method not only affirms the feasibility of noninvasive sonogenetics for targeting specific brain regions and modulating neuron activity but also opens new possibilities for the treatment of nervous-system-related diseases. However, it is important to note that while this approach directly activates the intended brain regions, it also inadvertently activates the auditory cortex (AC) and potentially a broader range of cortical areas. Therefore, further investigation is warranted to thoroughly assess the effectiveness of sonogenetics when targeting specific regions in the future. In addition, exogenous nanoparticle-based acoustic stimulation techniques have also been used in optimization studies of other biological techniques, such as imaging and cell sorting. Imaging technology based on acoustic waves (especially ultrasound) can compensate for the existing disadvantages of relying on genetically encoded fluorescent indicators for imaging, such as the narrow imaging range due to the limited tissue penetration of visible light and the single-pass nature of immunofluorescence staining. Wu et al. developed genetically encoded ultrasonic actuator gas vesicles (GVs), a unique class of air-filled protein nanostructures naturally produced in buoyant microbes, on a submicron scale [67]. These gas-filled protein nanostructures can significantly enhance the cellular response to the force of acoustic radiation. Furthermore, the study achieved the selective acoustic sorting of mammalian cells by transfecting these GVs into Human Embryonic Kidney (HEK) 293T cells. The development of similar synthetic biomolecules can facilitate the exploration of cellular and even molecular mechanisms in a wider field. Biological technology based on acoustic effects is expanding.

### 2.4. Magnetogenetics: Steering Neural Activities with Magnetic Fields

Magnetogenetics is the activation of specific ion channel proteins (such as TRPV1 or transient receptor potential vanilloid 4 (TRPV4)) by using the energy conversion caused by a magnetic field acting on magnetic nanoparticles. The current methods of using magnetic nanoparticles to regulate cell activity can be classified into two categories: (1) The mechanical force generated by the magnetic field on the magnetic nanoparticles on the cell surface activates the mechanically gated ion channels, acts on the cells, or activates specific receptors. (2) Temperature-sensitive ion channels are activated through the magnetocaloric effect of magnetic nanoparticles attached to the cell surface or gene-encoded endogenous metal nanoparticles, thereby regulating cell activity. A summary of studies on these two types of mechanisms is shown in Table 3. The following sections introduce the two mechanisms by which the magnetic field regulates cell activity.

#### 2.4.1. Regulation of Cell Activity by Mechanical Force Generated by Magnetic Nanoparticles

Mechanotransduction refers to the process of converting mechanical stimuli into electrochemical signals, which is closely related to regulating the conduction of proprioceptive signals and controlling blood flow velocity and direction [95]. Timely and accurate recognition and response of organisms to mechanical stimuli is crucial for organisms to adapt to ever-changing environments. Examples include ensuring normal growth and morphogenesis during embryonic development [106], maintaining the normal proliferation of cells in the body [107], and responding to sensory stimuli such as touch, hearing, and pressure. Studies have shown that there are specialized force-sensing cells in both plants and animals that respond to mechanical stimuli, and transmembrane ECM receptors are the key sites for the smooth transmission of forces into the cells [108]. The ECM can bind actin-associated proteins and connect to the microfilaments of the cytoskeleton. Therefore, in response to mechanical stimuli, the mechanical signals received by the microenvironment of the ECM can affect cellular activities, including cytoskeletal remodeling, migration, proliferation, differentiation, and intercellular communication. Some organisms have magnetic perception, just as birds use the Earth’s magnetic field to navigate. When organisms are placed in a magnetic field, the magnetic field is transformed into mechanical force or torque or acts by causing the aggregation of magnetically induced particles. At this time, the mechanosensitive gated channel existing on the cell membrane provides a medium for the cell to respond to the stimulation of the magnetic field. The magnetic field controls the opening or closing of the channel by activating the mechanically gated ion channel, thereby stimulating or inhibiting the activity of the cell or causing specific receptors to aggregate to activate cell signal transduction.

Inspired by the above phenomena, researchers have tried to use magnetic fields to act on organisms to manipulate cell activity, but the weak interaction force with biomolecules necessitates magnetic nanoparticles as intermediaries [101]. Nanoparticles are materials with at least one of three dimensions on the nanometer scale, which have been developed and widely used in the preparation of biosensing tools, the development of health care products, and imaging [101]. The unique chemical properties and crystal structure of nanomaterials compared with ordinary materials determine their unique characteristics in sensing light, electricity, force, and magnetism. For example, plasma-membrane-targeted gold nanorods (pm-AuNRs) synthesized from cationic protein/lipid complexes can activate the thermosensitive cation channel TRPV1 in neuronal cells by inducing a localized highly photothermal effect [109]. Compared with gold nanorods coated with traditional synthetic polymers, pm-AuNRs can minimize cell membrane damage and increase the possibility of developing novel targeted phototherapy technologies. Nanomaterials that can sense magnetism are called magnetic nanoparticles. Since Crick and Huges first proposed in 1950 that micron-sized magnetic beads can exert a certain force on organisms through magnetic field induction, an increasing number of studies have begun to focus on the role of mechanical forces under magnetic stimulation in changing cellular properties [110]. To date, in the field of biomedical engineering, magnetic nanoparticles have been used for (1) cell labeling and tracking and bioseparation; (2) targeted drug delivery and delivery of genes and radionuclides; (3) changing cell properties to form artificial blood vessels; (4) catabolizing tumor tissue and electromagnetic hyperthermia for cancer treatment; and (5) magnetic resonance imaging and other fields. Existing evidence combined with the properties of special electromagnetic effects at the micro/nano scale of magnetic nanoparticles suggests that magnetic nanoparticles may be an effective medium for in vivo and in vitro cell activity and neural signal transmission induced by electromagnetic fields.

Magnetic nanoparticles are a core–shell structure composed of a magnetic core and a polymer shell [111]. They have the advantages of magnetic responsiveness, small size effects, and biodegradability. They can achieve directional movement under the mechanical force generated by an external magnetic field. Magnetic nanoparticles can be prepared by the precipitation method, thermal decomposition method, microemulsion method, and sol–gel method. At present, a variety of optimized structures of magnetic nanoparticles have been designed to respond to magnetic fields and further regulate cells through mechanical effects. Since bare magnetic nanoparticles are biologically toxic when reaching a certain concentration, current research often chooses to coat the surface of the nanoparticles with a biocompatible polymer (e.g., polyethylene glycol) to reduce the biological toxicity. Early studies have shown that magnetic distortion technology can change the skeleton structure of cells, thereby activating MSCs (for example, Ca^2+^ ion channels) [112]. This effect is commonly found in cells of a wide range of tissues, including bone and muscle [113]. In 2008, Mannix et al. first proposed using 30 nm superparamagnetic nanobeads coated with monovalent ligands to bind to transmembrane receptors and expose them to an external magnetic field to make them magnetized. At this time, the transmembrane receptors aggregate due to the mechanical effect mediated by nanobeads and further open the biochemical signal transduction pathway, converting the magnetic input into the physiological signal of the cell for output [104]. Subsequently, more studies have confirmed the important role of the mechanical force generated by magnetic nanoparticles in the process of inducing magnetic fields in regulating cell activity and ion channel switching. For example, the application of an applied magnetic field (0.2 T, 2 h) can polymerize targeting antibodies to DLD-1 colon cancer cell death receptor 4 coupled to zinc-doped iron oxide magnetic nanoparticles (Zn_0.4_Fe_2.6_O_4_), which, in turn, induces apoptosis in zebrafish cells by activating signaling cascades such as caspase-3 [103]. The extent of apoptosis is proportional to the strength and duration of the magnetic field, independent of its direction [103].

While magnetic nanoparticles are widely used in biomedical research, their temporal resolution and targeting remain challenges, requiring size and structure optimization for precise regulation. The latest research has made some improvements to the above problems. For example, monovalent targeted magnetoplasmonic nanoparticles can achieve single-cell and molecular-level positioning and mechanical targeted delivery through unique ligand-receptor binding, programmed spatial reorganization, and targeted delivery of mechanical signals. The general applicability of magnetic nanoparticles was confirmed by studying the mechanical response of two important membrane proteins involved in intercellular communication (Notch and VE-calmodulin) in the presence of magnetic fields [94]. In addition, cubic magnetic nanoparticles can precisely and rapidly control the position of the static cilia of inner-ear hair cells by binding to endogenous glycoproteins of the membrane and displacing them by tens of nanometers under the action of an applied magnetic field (e.g., electromagnet) [95]. This magnetically gated switch realizes the control of cell activity at the single-cell level with high temporal resolution from seconds to microseconds, which provides the possibility of high temporal precision comparable to that of optogenetics. Such a modular nanoprobe system comprises several essential components: a Zn-doped ferrite core, a plasmonic shell, and an oligonucleotide-based targeting module. Through precise tuning of the modular characteristics of these nanoprobes—such as adjusting the probe’s size, mass, valence, and the structural domains within its targeting module—modular nanoprobes effectively address challenges related to target accessibility, receptor diffusion kinetics, and the detection of specific stresses acting on individual receptors. This tailored approach enhances the effectiveness of magnetogenetic modulation in regulating cellular activity.

In conclusion, the regulation of cell activity by the mechanical force generated by magnetic nanoparticles is one of the important regulatory mechanisms of magnetogenetics. Existing studies have aimed at different levels of regulation of single-cell behavior mediated by magnetic nanoparticles, including ion flow, protein migration, cell growth, development, and differentiation [98,100]. In addition, current research is also focused on optimizing the characteristics of magnetic nanoparticles, such as size, quality, and surface coating structure, to lay the foundation for the construction of new neuromodulation technology with high spatial and temporal resolution. In the future, the following four aspects should be considered and optimized: (1) to explore safe and effective magnetic field strength, stimulation time, and direction of magnetization; (2) to construct magnetic nanoparticles with high spatiotemporal precision and high targeting; (3) to clarify the potential mechanisms of the magnetic effect mediated by magnetic nanoparticles in the process of inducing mechanical force; and (4) to clarify the application of magnetogenetic technology to activate or inhibit the potential changes in cells at the cellular or even molecular level.

#### 2.4.2. Modulation of Cell Activity Through the Magnetocaloric Effect of Magnetic Nanoparticles

Magnetic nanoparticles provide innovative methods for neuromodulation through their unique effects. This section explores two primary mechanisms for regulating cell activity: the magnetocaloric effect of magnetic nanoparticles attached to the cell surface and modulation through genetically encoded endogenous metal nanoparticles. These complementary approaches not only enhance our understanding of magnetic-nanoparticle-based neuromodulation but also provide versatile tools for future research.

##### Regulation of Cell Activity Through the Magnetocaloric Effect of Magnetic Nanoparticles Attached to the Cell Surface

The changing magnetic field excites the electric field in space, and the changing electric field then excites the magnetic field, with the alternating excitement leading to both fields propagating into the distance from both directions, generating electromagnetic waves. Electromagnetic waves possess the unique capability to penetrate biological tissues without being detected by organisms. Additionally, they can be radiated over considerable distances to specific targets within a defined space. This theoretical capacity makes them an appealing physical stimulus for the remote and noncontact neuromodulation of biological systems. However, it is important to note that proteins with the innate ability to sense and respond to electromagnetic waves have not yet been identified. Consequently, in current research on electromagnetic neuromodulation techniques, the approach often involves the local injection of exogenous substances, such as nanoparticles, to serve as the sensing medium for this purpose [114]. In addition to the ability of the magnetic field to exert mechanical forces on cells via magnetic nanoparticles to control cellular activity, the magnetic field is also able to modulate cellular activity through the magnetothermal effect of magnetic nanoparticles attached to the cell surface [90].

The earliest interest in the magnetothermal effects of electromagnetic waves originated in the study of hyperthermia therapy. “Hyperthermia therapy” is derived from the Greek word meaning overheating and is an ancient medical science [115]. Since the beginning of civilization, humans have known how to use heat to treat diseases. With the development of recent technology, especially the development of physical heating technology, electromagnetic wave hyperthermia therapy has gradually come into view. Its principle is to exploit the characteristic that the high-temperature lethal time of tumor tissues is much less than that of normal tissues; this is mainly used for tumor treatment [116]. Traditional hyperthermia therapy uses nanoparticle suspensions to promote target site heating, which may not provide sufficient energy for targeted heating because the heat generated by magnetic thermal stimulation will dissipate rapidly in water [91]. Therefore, the development of magnetic nanoparticles bound to cell membranes provides a better sensing medium to achieve magnetothermal neuromodulation.

Earlier studies have demonstrated that electromagnetic waves in the radiofrequency (RF) band can target cells expressing the temperature-sensitive ion channel TRPV1 via superparamagnetic manganese ferrite nanoparticles (MnFe_2_O_4_) to open the cellular ion channels by localized heating, which, in turn, induces calcium ion inward flow [84,90,117]. In these studies, the TRPV1 ion channels were genetically modified to be expressed in the target cells, ensuring a sufficient density of TRPV1 channels on the cell surface. This genetic modification is critical for achieving the desired neuromodulatory effects through localized heating. Such local heating is achieved by distributing magnetic nanoparticles in high density on the cell membrane surface. The heating effect of RF magnetic fields on cells is clearly limited to the area around the cell membrane, as demonstrated by temperature-dependent changes in fluorescence intensity following intracellular coexpression of DyLight549 synthetic nanoparticles and Golgi-targeted green fluorescent protein [90]. Specifically, after applying the RF magnetic field, only the fluorescence intensity of DyLight549 decreased, and the fluorescence intensity of the Golgi apparatus remained unchanged. To test the feasibility of radiofrequency magnetic fields to induce behavioral responses in living animals, the study also targeted magnetic nanoparticles coated with polyethylene glycol (PEG)-phospholipids to sensory neurons of *Caenorhabditis elegans*. Approximately 85% of the crawling activities of *Caenorhabditis elegans* stopped after receiving RF magnetic field stimulation for 5 s. This result preliminarily indicates that an RF magnetic field can activate the heat avoidance behavior of animals through the magnetocaloric effect mediated by magnetic nanoparticles [90]. Radiofrequency magnetic fields using nanoparticles show potential in controlling ion channels and neuronal signals, but targeting specific neurons and generalizing findings from nematodes to mammals remain challenges. Preclinical research on magnetothermal neuromodulation technology urgently needs to be carried out in mammalian groups.

In 2017, Munshis et al. successfully verified the effectiveness of superparamagnetic nanoparticles as an electromagnetic induction medium for cells expressing the temperature-sensitive ion channel TRPV1 in an alternating magnetic field [91]. They confirmed the feasibility of magnetogenetics-based neuromodulation techniques in awake, freely moving mice [91]. The magnetocaloric effect brought by the alternating magnetic field can activate ion channels through magnetic nanoparticles attached to the plasma membrane of TRPV1^+^ neurons, further activating neurons in the corresponding brain regions of the cortex and causing related behavioral changes. For example, activation of neurons in the motor cortex elicited circling behavior in mice; activation of the striatum elicited circling behavior; and activation of the ridge between the dorsal and ventral striatum elicited gait freezing. In conclusion, it has been shown to be possible to respond to the magnetocaloric effect of a magnetic field by implanting magnetic nanoparticles in specific brain regions of mammals and achieve brain neurostimulation of organisms without increasing the overall tissue temperature. This method overcomes the limitations of traditional optogenetics that require implants or tethered optical fibers to provide stimulation and solves the off-target effects of chemical genetics.

##### Regulation of Cell Activity Through the Magnetocaloric Effect of Genetically Encoded Endogenous Metal Nanoparticles

Studies have shown that functionalized magnetic nanoparticles can be targeted to specific cell surface markers (endogenous expression or exogenous gene introduction) and intracellular components, thereby achieving specific regulation of cell functions. However, the binding time of functionalized magnetic nanoparticles to targets in specific tissues needs to be further defined, and repeated injections are likely to be required to prolong the duration of action [118]. For certain cells, such as those in the central nervous system, repeated intracranial injections may limit the utility of these techniques. Therefore, it is necessary to use nontoxic/low-toxicity and nondegradable materials to stably and durably induce cellular responses without repeated injections. Therefore, continuous production of genetically encoded nanoparticles within target cells, which avoids repeated injections and achieves a durable response to magnetic fields, is an advantageous alternative [118].

Biological enzymes and other biological systems contain a variety of metals, and some organisms can also synthesize complex metal-containing structures from inorganic materials [119]. Although magnetite has been reported to exist in certain tissues in mammals, this finding is still controversial, and the route of synthesis is unclear. It is worth noting that nearly all mammalian cells contain various metals, with iron being particularly prevalent. Iron serves as a crucial component within the mitochondrial respiratory chain, highlighting its fundamental role in cellular processes. However, iron oxide undergoes the Fenton reaction and produces highly toxic metabolites, so iron is sequestered by ferritin in almost all organisms except yeast. Ferritin is a specialized protein that has a core of different molecular forms of iron surrounded by a protein shell [120]. Ferritin normally sequesters iron (II), preventing its conversion to iron (III) and generating free radicals. In the presence of iron, these proteins spontaneously form intracellular nanoparticles mainly composed of iron oxide [121]. This allows researchers to exploit the continuous production of magnetic nanoparticles genetically encoded within cells.

Studies have shown that cells overexpressing ferritin can be affected by magnetic fields [118]. The overexpression of viral and transgenic ferritin has found application as a contrast agent in magnetic resonance imaging (MRI) studies, both in vitro and in vivo. By overexpressing human ferritin heavy chain and divalent metal-ion transporter-1 (DMT1) in cells, these cells can be separated from unmodified cells using magnetic fields, leveraging their superparamagnetic properties for this purpose. This approach enhances the visibility and specificity of MRI, facilitating precise imaging and detection in various research and medical contexts [122]. Overexpression of ferritin chimeric peptide in HEK cells can move cells toward the magnetic field [84]. When intracellularly generated iron nanoparticles were targeted to TRPV1 by transfected or endogenous ferritin, ion flux and cellular viability were significantly altered when the cells were exposed to a magnetic field.

Cellular modulation using the magnetothermal effect of genetically encoded nanoparticles has been validated in several studies. Some of these studies used the modified multimodal cation channel TRPV1 [85]. The channel is a temperature-sensitive ion channel with an activation threshold of 42 °C, but it also responds to other stimuli, such as pH, chemical agonists (e.g., capsaicin) and other possible mechanical stimuli. TRPV1 can interact with ferritin in an indirect or direct manner. For example, TRPV1 forms chimeras with green fluorescent protein (GFP)-tagged ferritin via anti-GFP nanobodies [122]. Treatment of HEK cells, stem cells, or neuronal cell lines expressing the ferritin–TRVP1 complex by oscillation or static magnetic fields can cause an increase in intracellular calcium ions, phosphorylation of cyclic adenylate response element-binding proteins, and expression of c-Fos in a TRPV1-dependent manner [118]. In cell-attached patch clamp recordings, neurons were found to depolarize in magnetic fields and to exhibit increased action potential release rates in isolated brain slices, among other phenomena [123]. TRPV1 can be mutated into a chloride ion channel, which can inhibit neurons under the action of a magnetic field [118]. In vivo studies showed that oscillating or gradient magnetic field treatment of glucose-sensitive neurons in the ventromedial hypothalamus of mice expressing anti-GFP-TRPV1/GFP-ferritin resulted in elevated blood glucose and increased food intake in freely moving mice.

The activity of TRPV4 is also temperature-dependent, with an activation threshold of 34 °C, and the channel also responds to osmotic stimulation and agonist treatment [124]. In a system referred to as Magneto 2.0, a ferritin chimeric peptide is directly linked to the carboxy-terminus of a truncated version of TRPV4. Researchers have evaluated the efficacy of this system in modulating cellular activity in zebrafish, which resulted in abnormal behavior, as well as in mice. When this system was expressed in mice within reward-related striatal dopamine receptor 1-positive neurons, it exhibited a strong preference for areas that were subjected to magnetic fields. This innovative approach demonstrates potential in selectively influencing neuronal activity through the application of magnetic fields [88].

Inserting a ferritin-binding motif into the TRPV1/TRPV4 sequence allows for the binding of this channel to the cell’s endogenous ferritin. Huston et al. used this method to expose cells expressing the ferritin system to an RF electromagnetic field and observed calcium ion influx in the cells. The expression in chicken embryos exposed to magnetic fields successfully replicates heart and craniofacial malformations caused by elevated body temperature during pregnancy [89].

In summary, the gene-encoded endogenous magnetic-nanoparticle-mediated cell activity regulation system usually has three components: (1) a static or oscillatory magnetic field signal; (2) ferritin-coated iron nanoparticles; and (3) the modified multimodal ion channel TRPV1 or TRPV4. When a magnetic field is applied, energy is absorbed by the iron within the ferritin shell and transferred to an attached ion channel, which opens, allowing ions to enter the cell. The effectiveness of magnetogenetic regulation relies heavily on ion channel selectivity [118]. In this system, the influx of specific ions through engineered channels determines the cellular response—calcium entry typically leads to neuronal activation through membrane depolarization, while chloride flux often results in inhibition. This selective ion transport mechanism enables precise control over cellular activity in magnetogenetic applications, providing a powerful tool for targeted neuromodulation.

Applications of magnetogenetics have already begun to emerge, yet many questions continue to be elucidated (see Table 1). First, despite the demonstrated effectiveness of magnetic field stimulation, the precise mechanisms underlying the interaction between magnetic fields and the ion channel–ferritin complex have remained elusive. Theoretical calculations of the classical mechanism show that the mechanical force or heat generated by a static or oscillating magnetic field acting on ferritin is not sufficient to activate the ion channel. However, recent theoretical work supported by experimental results proposes an alternative mechanism based on the magnetocaloric effect. Duret et al. proposed that in the absence of a magnetic field, the magnetic moments in ferritin are arranged in a random manner and thus have a high magnetic entropy [88]. However, in the presence of a magnetic field, the magnetic moment aligns with the magnetic field, reducing the magnetic entropy. To compensate for the loss of magnetic entropy, molecular vibrations increase, and heat is generated. The authors suggest that such a temperature change is sufficient to increase the probability that TRPV1 or TRPV4 channels will open and will open approximately one-tenth of the total number of channels, a proportion of channels sufficient to cause detectable calcium inward flow and depolarization in neurons or HEK cells. In addition, the oxidative stress response of cells in alternating magnetic fields is one of the possible chemical mechanisms that mediate the interaction between magnetic fields and ion channels. Recent investigations into the chemical mechanisms mediating magnetic field interactions with ion channels have revealed an important role for oxidative stress responses [125]. The inhibition of reactive oxygen species (ROS) generation has been shown to suppress the magnetic-field-induced activation of TRPV1 channels. This finding suggests that ROS production, particularly around ferritin nanoparticles, may serve as a crucial intermediate step in the magnetogenetic signaling pathway, linking magnetic stimulation to channel activation [125].

Second, noninvasive modulation of neural activity with magnetic fields via the cell surface or endogenous nanoparticles can theoretically modulate multiple locations in the loop without the need for implants. The main obstacle to implementing this method in practical applications is the introduction of the constructs (media) into the cells and the generation of a sufficiently strong magnetic field to activate the cells. However, although the durability of these external particles is inconclusive, as mentioned previously, further work is needed to determine the binding time of these exogenous particles to the cell membrane.

Third, although the harmful response of nanoscale magnetic stimulation to organisms may be weak under long-term exposure conditions, other magnetic field strengths still have potential harmful effects, such as tissue heating and damage. However, there is also the belief that in rodents, moderate local heating (<2 °C) does not alter or inhibit neuronal activity in different brain regions [126]. An increase in cortical temperature of more than 2 °C leads to abnormal firing in some cell types, while an increase of more than 4 °C leads to tissue damage [127]. It has been reported that a temperature increase to 42 °C causes thermal damage to brain tissue in 25 min [128]. At 43.3 °C, there was no sign of increased apoptosis or activated microglia [14]. The above arguments suggest that how to balance regulatory efficiency and tissue thermal damage is a problem that needs attention in future research.

Finally, in the field of neuromodulation, static magnetic fields and RF fields represent significant advances in minimally invasive approaches, offering the ability to modulate neural activity through intact tissue without surgical intervention. However, these techniques face practical limitations in clinical applications, primarily due to the requirement for external hardware such as resonant coils positioned near the subject’s head. While this approach effectively avoids tissue damage and the complications associated with chronic implants, it introduces constraints on subject mobility that are particularly challenging when transitioning from animal studies to human applications. These limitations necessitate careful consideration in the design and implementation of clinical protocols.

## 3. Non-Genetic Physical Factor Neuromodulation Technologies

Physical approaches based on non-genetic brain stimulation include deep brain stimulation (DBS), transcranial magnetic stimulation (TMS), transcranial direct current stimulation (tDCS), transcranial ultrasound stimulation (TUS), photobiomodulation therapy (PBMT), infrared neuromodulation (INM), electromagnetic stimulation (EMT), sensory stimulation therapy, and multi-physical-factor stimulation techniques (see Figure 2). Thus, the following is a brief review of several non-genetic physical neuromodulation techniques to compare them with genetic neuromodulation techniques and provide a reference for future specific treatment indications.

### 3.1. Deep Brain Stimulation: Targeting the Core

DBS is an established neuromodulation technique that utilizes precisely positioned bipolar electrodes—consisting of both positive and negative contacts within a single lead—surgically implanted in specific brain regions.

In DBS treatment, the two electrodes can induce an extracellular current and generate an extracellular electric field that induces transmembrane current; when the membrane capacitance reaches the threshold, the opening and closing of voltage-sensitive ion channels will be affected, regulating the excitation and inhibition of cells [131]. It is generally believed that modern DBS was developed in 1987. The team of the neurosurgeon Benabid and the neurologist Pollak reported for the first time that DBS to the subthalamic nucleus caused significant improvement [132]. In 2009, the US FDA approved the use of DBS for the treatment of refractory obsessive–compulsive disorder [133]. A series of studies also confirmed the important role of DBS in alleviating depressive symptoms and improving abnormal cognitive performance of patients [134,135]. To date, this technology has been approved by the FDA for clinical treatment of PD, obsessive–compulsive disorder, dystonia, and epilepsy [136,137]. Existing studies have launched multiple clinical trials with different targets, including the subcallosal cingulate gyrus (SCG) [134], ventral internal capsule and ventral striatum (VC/VS) [138], and nucleus accumbens (NAcc) [135]. For example, the study found that after 6 months of SCG-DBS treatment, patients with major depressive disorder and bipolar depression had a significant decrease in Hamilton Depression Scale scores, and the symptom remission rate reached 45% [134]. Long-term (12-month) DBS treatment showed a nearly 50% decrease in depression scale scores, and patients reported good social function recovery [139]. Clinical studies focusing on patients with treatment-resistant depression (TRD)—a severe form of major depressive disorder that fails to respond to at least two different antidepressant treatments—have shown promising results with targeted neuromodulation approaches. Specifically, deep brain stimulation of the nucleus accumbens (NAcc-DBS), a key brain region involved in reward processing and emotional regulation, has demonstrated significant therapeutic potential. After 12 months of NAcc-DBS treatment, patients showed marked improvement in anxiety symptoms [135]. The nucleus accumbens, part of the ventral striatum, serves as a critical interface between motivation, pleasure, and action, making it a strategic target for treating depression. This treatment approach is particularly significant for TRD patients who have exhausted conventional therapeutic options, offering a new avenue for symptom relief and improved quality of life.

The efficacy of DBS has been widely recognized, but DBS technology has some obvious disadvantages. For example, the implementation of DBS requires the expertise to have a high level of surgical operation; as a long-term treatment, the parameters need to be adjusted for each stimulation to match the best characteristics of disease progression. The implanted electrodes will also cause permanent damage to the brain tissue and the covering skull/scalp and at the same time induce a chronic immune response at the implant–tissue interface, which also brings certain obstacles to clinical promotion. The temporal interference (TI) DBS technology reported in the recent literature can activate neurons in specific deep brain regions without affecting the neuron activity of the overlying cortex without electrodes, which overcomes the problems of invasiveness and spatial resolution to a certain extent [140]. However, the stimulation equipment of the above method needs to be close to the body surface, which greatly limits the free movement of the experimental/treatment subjects. Aiming at the inconvenience of adjusting stimulation parameters, closed-loop neurostimulators provide a feasible solution. In 2013, the American company NeuroPace pioneered responsive neurostimulation (RNS^®^), which was approved by the FDA for the treatment of drug-refractory epilepsy. In 2021, the brain–computer interface clinical translational research team of Zhejiang University implanted a self-developed closed-loop neurostimulator into the brain of an epilepsy patient with a 19-year medical history. Compared with the condition before the device was implanted (or the stimulation system was turned off) (more than twenty attacks in a month), the patient only had one attack in a month [141]. In addition, the optimization of DBS technology also involves the extension of battery life and the reduction in side effects. In the future, DBS is expected to become one of the most mature means of neuromodulation.

### 3.2. Transcranial Magnetic Stimulation: Magnetic Pulses to the Cortex

TMS is a noninvasive brain stimulation technology that induces local currents in the cerebral cortex through strong pulsed magnetic fields to regulate neuron electrical activity and electrical signal transmission [142]. As early as 1896, the French doctor and physicist Arsonval thought of using a magnetic field coil to stimulate the human head to induce visual hallucinations. In 1985, Professor Anthony Barker of the University of Sheffield in the UK invented the first modern transcranial magnetic stimulator. In 2008, the FDA approved the first transcranial magnetic stimulation product NeuroStar (Neuronetics, Malvern, PA, USA, 510k number: K083538). Successively, Nexstim Navigated Brain Therapy (NBT) System 2 (Nexstim, Helsinki, Finland, 2012, 510k number: K182700), Brainsway Deep TMS System (Brainsway, Jerusalem, Israel, 2013, 510k number: K122288), Magstim Horizon 3.0 TMS Therapy System (Magstim, Whitland, UK, 2015, 510k number: K211389), MagVenture TMS Therapy (MagVenture, Farum, Denmark, 2015, 510k number: K193006), Neurosoft TMS (Neurosoft, Moscow, Russia, 2016, 510k number: K160309), and Apollo TMS Therapy System (MAG & More, München, Germany, 2018, 510k number: K180313) have also been approved by the FDA and put into use.

At present, in addition to directly regulating cortical function, TMS can further regulate subcortical and transcortical regions connected to cortical regions through neuron connections between brain regions, induce excitation and inhibition of brain regions, and change various behavioral performances. According to the stimulation mode, TMS is divided into single-pulse TMS, double-pulse TMS, and repetitive TMS (rTMS) [143]. Among them, single-pulse and double-pulse TMS are often used to explore brain function, while rTMS is often used to induce changes in brain activity and remodel damaged or abnormal neural circuits [144]. According to the stimulation frequency, rTMS can generally be divided into low-frequency rTMS and high-frequency rTMS. Low-frequency rTMS (≤1 Hz) reduces local cortical excitability, while high-frequency rTMS (≥5 Hz) increases local resting cortical excitability [145]. To date, TMS has been approved by the FDA for the treatment of neurological and mood disorders such as depression, obsessive–compulsive disorder (OCD), and migraine [146]. This method can also be used to aid in smoking cessation when patients are refractory to standard treatments [147]. Currently, the application of rTMS in the treatment of epilepsy, stroke, HD, and other diseases is actively expanding [148,149,150]. To date, many clinical trials have verified the beneficial effect of rTMS in the treatment of neuropsychiatric and other diseases. For example, OCD is a psychiatric disorder characterized by persistent, intrusive obsessions, and the medial prefrontal cortex, anterolateral orbital frontal cortex (OFC), and dorsal anterior cingulate cortex (dACC) are closely involved. Low-frequency rTMS stimulation of the mPFC and dACC in patients with obsessive–compulsive disorder can reduce the Yale–Brown Obsessive–Compulsive Scale (Y-BOCS) score, suggesting improved obsessive–compulsive symptoms [151]. A relatively mature rTMS treatment model has been developed for the treatment of depression. The current FDA-approved rTMS treatment mode is as follows: 10 Hz stimulation for 19–37.5 min per treatment at 120% of the resting motion threshold, 20–30 times a course, 5 times/week for 4–6 weeks; usually, a duration of 26–28 min produces the best treatment effects [152]. Compared with healthy people, patients with sleep disorders often exhibit abnormal hyperactivity of the dorsolateral prefrontal cortex (dlPFC). High-frequency rTMS treatment to the left and/or right dlPFC can effectively reduce cortical excitability and alleviate the symptoms of sleep disorders such as primary insomnia [148]. Davis et al. (2016) reported a case involving a 77-year-old male patient with late-onset HD, TRD, and generalized anxiety disorder (GAD) who underwent deep repetitive transcranial magnetic stimulation (dTMS) [153]. In this case, the patient received dTMS treatment daily for 49 days. The treatment parameters were set at 1600 pulses, with a frequency of 1 Hz, and an intensity at 120% of the motor threshold, utilizing an H-coil targeting the right dorsolateral prefrontal cortex. Notably, the patient maintained symptom relief for eight months post-treatment without requiring maintenance therapy. Additionally, the patient self-reported improvements in cognitive impairment, anxiety, and physical pain. While the patient experienced side effects such as tearing in the right eye and scalp discomfort at the treatment site, these adverse effects did not detract from the overall treatment efficacy. This case study highlights the potential efficacy of dTMS as a noninvasive neuromodulation technique in ameliorating symptoms in patients with late-onset HD, TRD, and GAD, providing a new perspective for the treatment of neuropsychiatric disorders [150]. The wide application of rTMS and its good curative effects have been widely confirmed.

TMS technology has the advantages of being painless, noninvasive, easy to perform, safe, and reliable. However, the specific mechanisms by which rTMS induces changes in neural activity remains unclear. Numerous studies have shown that rTMS regulates synaptic long-term potentiation and long-term depression [154], neurotransmitter levels [155], ion channels [156], and plasticity-related gene expression [157] to improve brain neuroplasticity and promote the reactivation of classical brain networks and recruitment of compensation networks, thereby promoting functional reorganization and recovery. The major drawbacks of TMS are that it cannot be focused on specific brain areas, and its spatial accuracy is rough. It has also been suggested that this therapy may be state-dependent. For example, cocaine users did not exhibit beneficial effects of TMS neuromodulation compared with healthy controls [158]. In the future, it is necessary to continue to optimize the parameters of rTMS and precisely target brain regions to improve the therapeutic effect, combining equipment such as electroencephalography (EEG), electromyography (EMG), or functional magnetic resonance imaging (fMRI) in the diagnosis and treatment of diseases for more extensive and precise treatment applications.

### 3.3. Transcranial Direct Current Stimulation: Steady Currents for Focused Effects

tDCS operates through a precise configuration of two or more electrodes placed strategically on the scalp, delivering controlled weak electrical currents (typically 1–2 mA, not exceeding 4 mA) to modulate neural activity [159]. The mechanisms of action can be categorized into three temporal phases: (1) Immediate effects involve altering neuronal membrane potentials, with anodal stimulation generally increasing neural excitability and cathodal stimulation typically decreasing it. (2) Intermediate effects involve the modulation of glutamatergic and GABAergic neurotransmission, affecting N-methyl-D-aspartate (NMDA) receptor efficiency and intracellular calcium levels. (3) Long-term effects emerge through the induction of synaptic plasticity via long-term potentiation (LTP) and depression (LTD)-like mechanisms. The resulting changes in neural network dynamics and connectivity patterns can persist beyond the stimulation period.

Long-term tDCS treatment can also lead to several persistent changes in neuronal excitability or increase neuronal plasticity by modulating the subthreshold of neurons’ resting membrane potential; increasing blood flow velocity and blood oxygen saturation and regulating changes in regional cerebral blood flow and cerebral blood volume [160]; modulating connections between ipsilateral and contralateral brain networks of a lesion by unilateral stimulation of specific brain regions [161]; and activating the level of neurotransmitters in the brain, such as endorphins, to improve the abnormal mental representation of patients [162]. Based on these mechanisms of action, tDCS is widely applied in the treatment of various conditions, including Alzheimer’s disease (AD) [163] and traumatic brain injury [164]. In 2020, the International College of Neuropsychopharmacology (CINP) released the latest clinical guidelines for the application of tDCS in neurological and psychiatric diseases, proposing nine indications, stimulation sites, and inapplicable conditions for tDCS treatment [165]. Studies of tDCS treatment in patients with severe depression demonstrate that tDCS can increase the excitability of the left dlPFC, effectively regulate the brain’s emotion-related circuits, and relieve depressive symptoms [165].

Compared with other neuromodulation technologies based on physical factors, tDCS has more obvious advantages. It offers noninvasive regulation, good tolerance, and the possibility for patients to be treated at home, thus covering a wider range of treatment groups [166,167,168]. However, despite these advantages, tDCS also faces significant limitations. Firstly, its spatial accuracy is poor, as it often cannot precisely target specific brain regions or local tissues. This is a notable drawback compared to other neuromodulation techniques such as DBS or TMS [169]. Additionally, the parameters of tDCS may need to evolve with disease progression, necessitating immediate adjustments and substantial technical expertise from the operator [170]. Furthermore, the electric current can produce a thermal effect on biological tissue, making it crucial to establish safe use parameters to achieve optimal stimulation effects efficiently [171]. Another important limitation is the potential for adverse skin reactions, such as mild tingling and slight itching under the stimulation electrodes [172]. Although rare, there have been reports of more severe side effects, such as epilepsy-like symptoms of respiratory and motor paralysis during treatment. While the causality of these severe reactions remains unclear, advancements in equipment and techniques have likely mitigated such risks.

Meta-analyses have indicated that despite its use in treating AD, the efficacy of tDCS is inconsistent. The variability in study outcomes is primarily due to several critical factors, including the heterogeneity of stimulation parameters, cognitive measures, and evaluation procedures [173]. For example, a large interval between assessment and stimulation can weaken the results [174], and a lack of randomized or blinded clinical trials often leads to an overestimation of efficacy [163]. These factors hinder definitive conclusions about the effectiveness of tDCS in achieving stable behavioral changes [163,175].

In conclusion, while tDCS holds promise due to its noninvasive nature and potential for home treatment, standardizing parameters and conducting high-quality studies are essential steps towards making tDCS a viable treatment option for several neurological disorders. Future research should focus on optimizing stimulation protocols, enhancing targeting accuracy, and minimizing side effects to fully harness the therapeutic potential of tDCS.

### 3.4. Transcranial Ultrasound Stimulation: Acoustic Precision

Before introducing TUS, it is essential to clarify the key differences between TUS, a non-genetic neuromodulation technology, and sonogenetics (see Section 2.3 for details). Sonogenetic and non-genetic sonic stimulation techniques both utilize ultrasound waves to achieve neuromodulation, but they differ fundamentally in their approach and application. Sonogenetic stimulation involves the genetic modification of cells to express ultrasound-sensitive proteins, such as TRPA1 or Piezo1, allowing for precise control of neuronal activity. In contrast, non-genetic sonic stimulation, like TUS, leverages the mechanical and thermal effects of ultrasound waves to modulate neuronal activity directly, making it suitable for clinical applications due to its noninvasive nature.

TUS is a noninvasive neuromodulation technique that passes ultrasound waves through the skull to reach specific areas of the brain to affect neuronal activity and brain function. This technology stimulates brain regions with focused ultrasound energy, has a deeper penetration range and higher focus resolution, and shows far-reaching application prospects in the treatment of neuropsychiatric diseases [176]. As early as 1939, Reimar Pohlmann first tried to use ultrasound to treat sciatica. However, significant progress was made in the 1950s by Professor Fry’s team from the Bioacoustics Laboratory of the University of Illinois at Urbana, who first applied ultrasound to animal nerve tissue stimulation. Early studies showed that ultrasound can pinpoint lesions down to the millimeter, enabling localized stimulation without damage to surrounding nontarget tissue [177]. Subsequently, a team completed the first focused ultrasound neuromodulation of the primary visual cortex in a cat in 1958 and predicted that this technology had great potential in the treatment of neuropsychiatric diseases [178]. Since then, ultrasound has been widely studied as an important neuromodulation technology, especially in the past ten years. TUS not only induces behavioral changes but also affects the activity of individual neurons [179], local field potential changes [180], and EEG signal changes [71]. Furthermore, prolonged ultrasound stimulation has been shown to induce changes in neural activity for up to 2 h [181].

Several major hypotheses exist regarding the mechanisms of action of TUS, including the thermal effect hypothesis, the mechanical effect hypothesis, the altered membrane permeability hypothesis, and the neurotransmitter modulation hypothesis. Although some progress has been made, the exact mechanisms still require further study. The thermal effect hypothesis suggests that ultrasound can cause local thermal effects and affect neuronal excitability and conductivity, but low-intensity ultrasonic parameters widely used in the field of neuromodulation can induce neuronal excitation and inhibition [182]. Therefore, the mechanism of action of high-intensity ultrasound may be dominated by thermal effects, while low-intensity ultrasound mainly relies on other effects. The mechanical effect hypothesis states that the mechanical vibrations of ultrasound may directly affect the electrical activity of neurons. The altered membrane permeability hypothesis proposes that ultrasound may temporarily alter cell membrane permeability, affecting neurotransmitter release and neuronal excitability. The neurotransmitter modulation hypothesis suggests that ultrasound stimulation may regulate neuronal activity by modulating neurotransmitter release and reuptake. Although some progress has been made, the mechanism of action still needs to be further studied, requiring a combination of experiments at different energy levels and techniques, such as neuroimaging.

Medium-intensity (100–200 W/cm^2^) and low-intensity (<100 W/cm^2^) focused ultrasound have different effects [62]. Moderate-intensity ultrasound is mainly responsible for breaking through the blood-brain barrier for drug delivery. LIFUS has a low transmission frequency and dose. The energy transmitted to the tissue is usually less than 3 W/cm^2^, which will not cause an excessive temperature increase at the target, thereby avoiding heat damage [183]. Therefore, LIFUS has been widely used in the fields of microparticle manipulation and neuromodulation. As an emerging ultrasound brain intervention technology, LIFUS has begun to be used in the treatment of neuropsychiatric diseases and neuroscience research in recent years. In July 2016, the FDA approved the use of TUS for refractory essential tremor [184]. Although clinical trials are still ongoing, TUS ablative and non-ablative therapy also showed positive effects in the remission of PD [185], epilepsy [186], and other disease symptoms. In addition, TUS has been suggested to have the potential to help regulate brain network connectivity, affecting higher functions such as recognition and memory [71].

A significant challenge in applying ultrasound brain intervention technology in humans is achieving precise transcranial focusing. The irregular shape of the skull and the inhomogeneity of sound velocity and density distribution can cause focal point shifts. This issue can be addressed through technological advancements such as sensors, microbubble incorporation, transducers, and electronic phase correction. This technology has demonstrated remarkable advantages in treating various neurological diseases. For instance, a recent study used oil as a coupling medium to leverage the hydrophobicity of the hair surface and induce blood–brain barrier opening, achieving high-quality acoustic coupling in unshaven mice [187]. This approach thoughtfully addresses patient concerns, particularly among female patients, about self-esteem and security related to hair removal during treatment, thereby advancing the humanization of clinical research.

### 3.5. Photobiomodulation Therapy and Infrared Neuromodulation: Light as a Healer

PBMT (also known as low-intensity laser therapy) is based on the mechanism of photobiomodulation and includes both visible and near-infrared light neuromodulation techniques. These methods use light to regulate neural activity, targeting and interacting with the inner membrane of mitochondria or other cellular structures, facilitating deeper tissue penetration and minimal invasiveness. When a low-level infrared spectrum laser is close to or in contact with the skin, the light energy can penetrate the skin tissue to reach and connect with the inner membrane of the mitochondria. The characteristics of the interaction with the cytochrome C complex trigger a series of biological cascade reactions. PBMT helps to increase cell metabolism, relieve pain, regulate the body’s immune response, promote tissue regeneration, improve damaged tissue microcirculation, and finally achieve the purpose of treatment. The efficacy and mechanism of action of PBMT have been extensively studied. The main mechanisms of PBMT include (1) improving brain microcirculation, increasing blood oxygen concentration and cerebral blood flow, and inducing a blood oxygen level-dependent (BOLD) response; (2) inducing nerve regeneration, assisting nerve enhancement, and exerting neuroprotective effects; and (3) activating various molecular pathways related to anti-inflammatory, anti-apoptotic, and antioxidant responses. For example, activation of the ERK/CREB signaling pathway mediated the upregulation of BDNF and increased cell survival [188], and activation of the Akt/GSK3β/β-catenin pathway reduced apoptosis [189]. The development of PBMT in the fields of damaged nerve rehabilitation, functional repair, and pain relief suggests that PBMT will show a wide range of benefits in the treatment of related diseases.

Infrared neuromodulation (INM) is a novel technique that directly stimulates neurons through temperature changes induced by infrared light, typically in the wavelength range of 1400–2100 nm [190,191]. INM operates through two primary mechanisms: infrared neural stimulation (INS) and infrared neural inhibition (INI) [190,192,193]. INS involves the excitation of neurons via the transient heating effects of infrared light, leading to depolarization of the neuronal membrane and generating action potentials [192], while INI inhibits neuronal activity by hyperpolarizing the membrane and activating potassium channels [194]. This noninvasive technique is versatile, modulating neuronal activity in both research and clinical settings, such as auditory restoration in cochlear implants [195], pain relief [196], muscle control [196], and treatment of neurological conditions like epilepsy and PD [195,197,198].

The field of INM is still evolving, with ongoing research aimed at better understanding the underlying biophysical mechanisms and improving the precision and safety of the technique. A recent study described the development of macromolecular infrared nanotransducers for DBS, which can absorb wide-field light in the second near-infrared spectral window (NIR-II, 1000–1700 nm) and control the temperature-sensitive ion channel TRPV1 for neuromodulation [193]. Other properties of the sensor have also been optimized, such as a polymer coating to ensure the water solubility and biocompatibility of the sensor and repeated near-infrared light stimulation to verify the stability of the sensor [193]. Advances in micro- and nano-engineered interfaces, such as polymer-mediated approaches, are expected to enhance the delivery and control of infrared light in neuromodulation applications [191,192]. In the future, combining INM with other technologies, such as MRI and electrophysiology, may provide more comprehensive approaches for studying and treating neurological disorders [192].

In conclusion, PBMT and INM have the advantages of being affordable and easy to perform at home, but there is also room for growth. For example, the optimal laser parameters (such as wavelength, laser power, and total energy) for treatment have not yet been clarified. Different wavelengths of laser light have different penetration depths in tissues, and different tissues have different absorption coefficients of light. For example, melanin has a strong ability to absorb light at lower wavelengths (500–800 nm), so when the target treatment site is dark skin, a laser with a longer wavelength should be used. The FDA divides lasers into four main categories (I-IV) according to output power and has approved Class IV lasers for joint pain treatment [199]. The higher the laser power is, the deeper the tissue depth that can be reached, but it can also cause damage to the eyes and other organs and even blindness in severe cases [200]. Therefore, when using a laser for PBMT treatment, special attention should be given to the selection of laser parameters during the treatment process. Second, the spatial and temporal resolution of PBMT still needs to be improved, and the regulation is limited to the superficial cortex, which greatly limits the application of this technology. In addition, the underlying mechanism of PBMT needs to be clarified, and the treatment cycle and treatment time of PBMT should be continuously explored to ensure the optimal therapeutic effect.

### 3.6. Electromagnetic Stimulation Therapy: Invisible Waves, Visible Change

The method in which electromagnetic fields act on the human body to treat diseases is called electromagnetic therapy. According to the energy of electromagnetic radiation, electromagnetic fields can be divided into ionizing radiation and nonionizing radiation. According to the frequency, the World Health Organization (WHO) further divides nonionizing radiation into static electromagnetic fields (0 Hz), extremely low-frequency magnetic fields (0–300 Hz), intermediate-frequency fields (300 Hz–10 MHz), and radiofrequency fields (10 MHz–300 GHz).

Radiofrequency therapy is a physical therapy technology that is applied in the medical field by introducing high-frequency electromagnetic field energy to produce tissue thermal effects and other biological effects. The thermal effect generates local heating through tissue resistance, leading to coagulation and denaturation of cell proteins, achieving cell coagulation and necrosis, and promoting tissue blood flow and metabolism, which is often used for cancer ablation [201]. Recent studies have also found that magnetic field therapy can accelerate bone healing by increasing mitochondrial oxidative phosphorylation (OxPhos) activity and promoting the bone-forming function of cells. Mice exposed to a 1.5 mT magnetic field exhibit better bone healing and higher bone density [202]. Other biological mechanisms also involve changes in cell membrane permeability and increased intracellular calcium ion concentration, which, in turn, affect cell function and tissue regeneration, but the specific biophysical mechanism has not yet been elucidated [203].

In recent years, in the field of neurological diseases, radiofrequency therapy has been shown to have potential curative effects in PD and epilepsy and can improve symptoms by regulating neuron activity [204,205]. Long-term radiofrequency electromagnetic therapy also prevents and reverses cognitive impairment in AD transgenic mice and even improves cognitive performance in normal mice [206]. Arendash et al. developed transcranial electromagnetic therapy (TEMT) for patients with AD [130,207]. Preliminary results showed that eight mild/moderate AD patients who underwent daily TEMT for two months showed a significant reversal in cognitive performance [207]. The results of recent pilot studies also suggest that long-term TEMT for 2–12 years can effectively alleviate the deterioration of AD symptoms, and the physiological and biochemical indicators show a significant increase in the levels of C-reactive protein; p-Tau217, Aβ1-40, and Aβ1-42 levels in cerebrospinal fluid decreased; and no safety issues occurred during treatment [130]. This study provides a feasible path for enriching and optimizing existing neuromodulation therapies. Zhi et al. found that spatial and working memory was improved in AD mice after long-term 900 MHz microwave exposure compared with sham exposure. Microwave radiation (900 MHz) for 180 or 270 days did not induce Aβ plaque formation in WT mice but inhibited Aβ accumulation in the cerebral cortex and hippocampus in 2- and 5-month-old APP/PS1 mice. This effect mainly occurred in the late stage of the disease and may have been attributed to the downregulation of apolipoprotein family members and SNCA expression as well as excitatory/inhibitory neurotransmitter rebalance in the hippocampus, suggesting that 900 MHz microwave exposure may be a potential therapy for AD.

Electromagnetic fields, particularly pulsed electromagnetic fields, have also demonstrated significant anti-inflammatory effects across various biological systems [208]. Studies indicate that PEMF exposure can reduce inflammation by modulating cellular pathways and receptors, notably by activating adenosine receptors, which downregulate pro-inflammatory cytokines such as interleukin (IL)-6 and IL-8 and promote anti-inflammatory factors like IL-10 [209]. Additionally, pulsed electromagnetic fields have been shown to decrease hypoxia-induced ROS production and cell apoptosis in neuron-like and microglial cells, partly by inhibiting hypoxia-inducible factor 1-alpha (HIF-1α) [209]. Clinically, pulsed electromagnetic fields have been used to promote cartilage repair [210], reduce postoperative pain [211], and enhance mesenchymal stem cell proliferation and differentiation [212], making them promising for treating conditions like osteoarthritis and neuroinflammation [213]. These findings suggest that pulsed electromagnetic fields offer a noninvasive and effective approach to mitigating inflammation, with potential applications in treating various inflammatory and degenerative diseases.

The above studies support the great potential of electromagnetic stimulation as an effective neuromodulation therapy in the treatment of diseases. However, there are still some problems in the existing research that need to be solved urgently. First, the underlying mechanism of EMT is unknown. The treatment of tumors and cancers is known to use high temperatures to kill cancer tissue cells, but the mechanism of improving cognitive ability remains to be explored. In addition, although radiofrequency therapy has the advantages of reaching deep brain regions and being able to operate at home, its time accuracy is low, and its efficacy needs to be verified by large-scale clinical trials.

### 3.7. Sensory Stimulation Therapy: Healing Through Engaging the Senses

Sensory stimulation therapy is the process of using sound, visible light, and other sensory inputs to modulate neural oscillations in the primary sensory cortices of the brain. These neural oscillations, which represent synchronized electrical activity patterns among large populations of neurons, are fundamental to information processing in the brain. For example, alpha waves (8–13 Hz) in the visual cortex help filter irrelevant visual information, while theta rhythms (4–8 Hz) in the AC are crucial for processing temporal aspects of sound. By delivering precisely timed sensory stimuli, this therapy can help restore or enhance these natural brain rhythms, thereby improving the patient’s cognitive function and behavior.

Studies have found that light and sound stimulation may be an effective method for treating cognitive impairment caused by sleep rhythm disorders. Bright light therapy has shown particularly promising results in Alzheimer’s patients, where it helps regulate the circadian rhythm and enhance cognitive performance [214]. In healthy older adults, exposure to blue-rich light at night has been demonstrated to improve cognitive performance the following morning, likely through its effects on circadian-regulated neural oscillations [215]. Short-term exposure to bright light may have beneficial effects on cognitive function in humans by increasing alertness-related brain rhythms in the cortex [216]. Low-level light therapy has become a major intervention for conditions such as seasonal affective disorder, sleep/wake disorder, and cognitive impairment [217,218]. The therapy works by stimulating photoreceptors in the retina, which then influence neural oscillations in both visual processing areas and deeper brain structures involved in mood and cognition. Additionally, dynamic single-sound stimuli delivered during slow-wave sleep can enhance the consolidation of verbal associative memories, possibly by reinforcing the natural delta wave (0.5–4 Hz) oscillations that characterize deep sleep and that are crucial for memory formation [219].

Professor Li-Huei Tsai’s team from the Department of Brain and Cognitive Sciences at the Massachusetts Institute of Technology developed a noninvasive therapeutic approach called Gamma ENtrainment Using Sensory stimulation (GENUS). GENUS uses precisely calibrated light or sound stimuli at a 40 Hz frequency to induce gamma oscillations in the brain, which can reduce amyloid accumulation in AD mouse models [220,221]. In their subsequent research, they used 40 Hz auditory tonal stimuli to drive gamma-frequency neural activity in both the AC and hippocampal CA1 region. After 7 days of auditory stimulation, they observed significant improvements in spatial and recognition memory, along with reduced amyloid plaque burden in both the primary auditory cortex and hippocampus of 5× FAD mouse models [222]. The treatment triggered significant activation of microglia and astrocytes and enhanced blood vessel function [222]. In a P301S tauopathy model, this auditory sensory stimulation was also effective in reducing levels of phosphorylated Tau [222]. The team further demonstrated that combining auditory and visual gamma-band sensory stimulation produced even more comprehensive benefits. This multi-sensory approach enhanced microglial responses and reduced amyloid accumulation in the medial prefrontal cortex. Whole-brain analyses revealed that this combined stimulation approach led to a broad reduction in amyloid plaques throughout the neocortex [222,223]. These findings suggest that GENUS represents a promising non-pharmacological intervention for AD, working through the modulation of brain oscillations to engage multiple cellular and molecular pathways involved in AD pathology.

Currently, sensory stimulation therapy as an innovative digital therapy for AD has achieved positive results in phase I/II (CA-0005) and phase II clinical trials (OVERTURE; NCT03556280) [224]. In a phase II study, 74 subjects were treated with the GammaSense (Cognito Therapeutics, Inc., Cambridge, MA, USA) stimulation system for one hour per day for six months [225]. The results showed that compared with the sham stimulation group, the 40 Hz γ-band “sound+vision” stimulation therapy significantly maintained the cognitive ability of patients; slowed the decline rate of cognitive function; and effectively guaranteed the safety and tolerability, compliance, and effectiveness of the therapy [225,226,227]. Recently, the team also extended sensory stimulation to tactile stimulation and used Tau P301S and CK-p25 mice (two neurodegenerative mouse models) to demonstrate that 40 Hz tactile stimulation can induce primary motor neural activity in the primary somatosensory cortex, preventing neurons from dying or losing their synaptic circuit connections and reducing neuronal DNA damage. Both groups of mice subjected to 40 Hz vibrotactile stimulation showed improved motor performance [228]. However, there are still debates regarding the mechanism and efficacy of 40 Hz light stimulation, particularly its ability to modulate deep brain structure activity. While Tsai’s team demonstrated that 40 Hz auditory stimulation could effectively drive gamma-frequency neural activity in both the auditory cortex and hippocampal CA1 region, the effectiveness of light stimulation in engaging deep brain structures remains controversial. For example, the team of Professor György Buzsáki of New York University used APP/PS1 and 5× FAD mice as the research objects to explore the effects of acute and chronic 40 Hz light stimulation on Aβ deposition and glial cells in model mice. Compared with the control group, neither acute nor chronic phototherapy affected Aβ deposition or microglial morphology in the model mice [229]. This discrepancy might be attributed to the different penetration capabilities of light versus sound stimulation, as auditory pathways may provide more direct access to deeper brain structures. Therefore, while GENUS shows promise as a therapeutic approach, particularly with multi-modal stimulation, more mechanistic studies are needed to: (1) understand the differential effects of various sensory modalities on deep brain structures, (2) optimize stimulation parameters for maximum therapeutic benefit, and (3) identify the most effective combinations of sensory stimulation for comprehensive brain-wide effects.

Additionally, the above non-genetic techniques have shown significant potential in attenuating neuroinflammation and reducing the release of pro-inflammatory factors, thus offering additional therapeutic benefits. One of the primary beneficial effects of neuromodulation techniques lies in their sophisticated modulation of neuroinflammation, which encompasses the complex immune responses within the central nervous system. At its core, neuroinflammation involves the activation of microglia and the consequent production of pro-inflammatory cytokines, processes that can significantly impact brain health. These neuromodulation approaches demonstrate remarkable capability in regulating both central and peripheral immune responses through several elegant mechanisms: they can guide microglial cells to shift from their aggressive pro-inflammatory state toward a more neuroprotective profile, delicately adjust the blood–brain barrier’s selective permeability, and fine-tune the broader systemic inflammatory response that influences brain function. Techniques such as TMS have been shown to decrease the levels of pro-inflammatory cytokines in the brain. TMS, for instance, enhances the anti-inflammatory environment by modulating cortical excitability and plasticity, which reduces the release of pro-inflammatory factors and promotes neuroprotection [230,231]. Similarly, tDCS influences the production of neurotrophic factors and decreases pro-inflammatory cytokines, aiding in the treatment of neurodegenerative diseases [230,232]. Another significant benefit is the enhancement of neurotrophic factors. PBMT, using specific wavelengths of light, also contributes to the enhancement of neurotrophic factors. By improving mitochondrial function, PBMT reduces pro-inflammatory cytokine levels and promotes cellular repair and regeneration [233]. Additionally, the reduction in oxidative stress is a critical mechanism through which neuromodulation techniques exert their anti-inflammatory effects. TUS, for example, can noninvasively penetrate deep brain structures and modulate neuronal activity. It enhances the expression of anti-inflammatory cytokines and neurotrophic factors, thus reducing oxidative stress and neuroinflammation [230,234,235,236]. This makes TUS a promising tool for treating conditions like dementia and other neurodegenerative disorders. In conclusion, neuromodulation techniques offer substantial benefits in reducing neuroinflammation through the modulation of immune responses, enhancement of neurotrophic factors, and reduction in oxidative stress. These mechanisms highlight the therapeutic potential of neuromodulation in treating various neurological and psychiatric disorders. Continued research and technological advancements are essential to fully harness these benefits and develop more effective neuromodulation therapies.

### 3.8. Multi-Physical-Factor Stimulation Techniques

Given the limitations of temporal and spatial accuracy and the penetration depth of existing noninvasive neuromodulation methods with a single physical factor, researchers began to focus on stimulation therapy combining multiple physical factors. One such approach is magnetic–acoustic coupling stimulation, where magnetic fields work together with sound waves to achieve more precise brain stimulation—much like using two tools in concert to accomplish what neither could do alone. Another example is visual–electrical stimulation, which synchronizes light patterns that we can see with gentle electrical currents to enhance brain responses—similar to how a conductor might use both hand gestures and a baton to better guide an orchestra. These combined approaches aim to overcome the limitations of using any single method alone. In 2003, Stephen Norton proposed the possibility of using focused ultrasound in a static magnetic field, or transcranial magnetoacoustic stimulation (TMAS), as a new and potentially noninvasive brain stimulation technique for the treatment of neurological and mental illness [237,238]. Under the excitation of ultrasound, the charged particles (primarily ions like sodium and potassium) in the brain tissue are set in motion. When these moving charged particles encounter the magnetic field, they experience the Lorentz force—a fundamental physical force that pushes charged particles perpendicular to both their direction of motion and the magnetic field. This interaction creates brief electrical currents in the brain tissue, which can effectively influence neural activity in a highly controlled manner. According to Faraday’s law, the generated electric field is proportional to the velocity of the ion particles, making it possible to manipulate the stimulation effect. In addition, thanks to its small ultrasonic region and the ability of sound waves to maintain high energy transfer and low attenuation when penetrating tissues, TMAS can provide low-millimeter spatial resolution even in deep brain regions, and its focusing degree is 10 times that of TMS [239]. This gives TMAS an advantage in stimulating specific small-sized deep brain regions [237,238]. TMAS has been widely used in the exploratory treatment of neurological and psychiatric diseases [239,240]. For example, animal studies have shown that TMAS improves cognition in mice with PD [239]. The study found that compared with the control group, PD model mice treated with TMAS showed recovery in spatial learning and memory, which may be due to TMAS promoting the synaptic transmission of neurons in the hippocampus by increasing the expression of BDNF. Long-term potentiation or indirect regulation of cAMP response element-binding protein (CREB) affects the expression of brain-derived neurotrophic factor, thereby affecting the consolidation process of short-term and long-term memory [239]. TMAS can also shorten the reaction time of neural activity in the motor cortex and enhance neuromodulation [241].

Yuan Yi’s team from Yanshan University and Li Xiaoli’s team from Beijing Normal University also clarified the relationship between TMAS and the regulation of neuron activity and neural rhythmic oscillation activity [242]. This study deeply explores the mechanism of TMAS in the regulation of the nervous system and provides substantial theoretical support for its practical application in the field of neuroscience. The research results of Yuan Yi’s team emphasized the potential of TMAS to trigger the regulation of neuronal excitability and further revealed that it has a significant regulatory effect on neuron excitability by finely interfering with ion channel activity and membrane potential changes. This regulatory mechanism is expected to achieve precise and dynamic information transmission and integration in the local neural network, thereby affecting the functional status of higher-level brain regions. In-depth research on the complex relationship between TMAS, neuron activity, and nerve rhythm oscillation will help to fully grasp the regulatory effect of TMAS, further promote the development of this field, and lay a foundation for clinical practice.

Furthermore, the neuroengineering team at Tianjin University has pioneered a neurofeedback training approach that integrates visual stimuli and electrical stimulation. This innovative method has proven effective in enhancing the time-frequency characteristic response of EEG in subjects. It holds promising potential for application in clinical rehabilitation for individuals facing cognitive impairment, offering a novel avenue for improving cognitive function through advanced neurofeedback techniques [243].

## 4. The Road Ahead: Innovations and Integration

Despite notable progress, several challenges hinder the clinical translation of the above technologies. To achieve more successful clinical applications, it is essential to focus on key factors such as safety and efficacy, which are particularly important for genetic modification techniques like AAV injection and nanoparticle delivery, which require a thorough evaluation of potential side effects. Developing noninvasive or minimally invasive neural modulation techniques, such as ultrasound and magnetic stimulation, can enhance patient acceptance compared to traditional invasive methods. In addition, improvements to the operability and accessibility of these technologies, such as simplifying operational procedures and reducing reliance on specialized knowledge and complex equipment, could also facilitate broader application of these technologies.

However, the clinical translation of existing neuromodulation technologies presents numerous challenges. Traditional genetic neural modulation techniques like optogenetics and chemogenetics have been widely used in basic neuroscience research. Non-genetic neural modulation techniques based on physical factors have shown promising prospects in clinical treatment. However, current non-genetic neural modulation techniques tend to be invasive when regulating the activities of deep central nervous system structures, requiring permanent implants. This increases the risk of infection or fibrosis and may result in reduced efficacy over time. Moreover, invasiveness limits most implants to a single site or adjacent areas, necessitating specialized knowledge and complex equipment, which hinders widespread adoption. Translating genetic neural modulation from animal experiments to human applications may involve introducing, removing, or altering genetic material within patient cells to treat or prevent diseases. Emerging genetic neural modulation techniques using sound waves or magnetic stimulation, due to their noninvasive or minimally invasive nature, may be more acceptable to patients compared to implant-based treatments. However, challenges related to AAV injection for protein expression responsive to stimuli, nanoparticle safety, and appropriate stimulation dosage must be addressed. Evidence from bioimaging and cancer treatment indicates the potential cytotoxicity of certain magnetic nanoparticles, which can cause cellular damage by inducing oxidative stress, activating reactive oxygen species, promoting inflammatory responses, and causing DNA damage [244]. These challenges suggest that genetic neural modulation techniques still have some way to go before clinical translation.

Future research should focus on optimizing delivery methods, such as developing more efficient and safer AAV and nanoparticle delivery systems. For instance, AAVs that can be delivered to the central nervous system via peripheral injection, targeting deep brain structures using specific viral capsid proteins or promoters to target specific cell types, lay the foundation for (micro)invasive neural modulation applications. Although current delivery methods still require injections, technological advances and the introduction of new therapies will support innovative delivery methods like aerosol therapy. Ensuring the safety of genetic neural modulation technologies is paramount. This includes optimizing and evaluating the toxicity of magnetic nanoparticles in vivo and in vitro to ensure their safety in cellular activity regulation. Comprehensive safety assessments of applied ultrasound and electromagnetic fields are also necessary. Leveraging emerging technologies such as CRISPR gene editing can further validate their effectiveness in neural modulation [245]. Recent preclinical studies using tools like CRISPR have shown promising potential. Additionally, the complementary integration of multiple-physical-factor modulation methods can address the limitations of existing single-neural-modulation approaches, leading to more significant clinical disease treatment outcomes.

In conclusion, although current neural modulation technologies are at the forefront of scientific research and some methods have been clinically applied, emerging candidate methods continue to evolve, enhancing and complementing existing neural modulation technologies in terms of modulation distance, speed, signal-to-noise ratio, compatibility with free behavior, and minimal invasiveness. It is crucial to bridge gaps and foster technological innovations to overcome current limitations and fully harness the therapeutic potential of neural modulation technologies. Addressing these challenges will pave the way for more effective and widely accessible neural modulation therapies, ultimately improving the prognosis for patients with neurological and psychiatric disorders.

## Figures and Tables

**Figure 1 cells-14-00122-f001:**
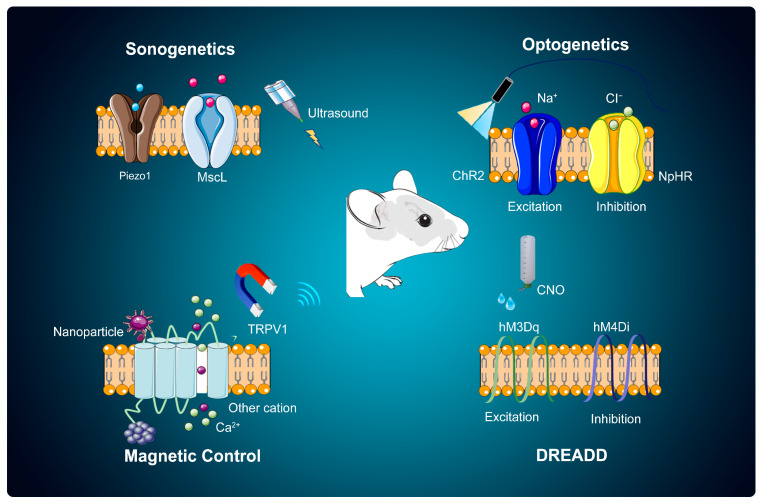
Schematic diagram of genetic neuromodulation technologies. Genetic neuromodulation encompasses sonogenetics, optogenetics, magnetogenetics, and chemogenetics. Sonogenetics employs ultrasonic stimulation to activate mechanosensitive ion channels, such as Piezo1 and mechanosensitive channel of large conductance (MSCL), inducing neuronal activity. Optogenetics utilizes blue light to stimulate the excitatory ion channel channelrhodopsin-2 (ChR2) and yellow light for the inhibitory ion channel natronomonas halorhodopsin (NpHR), causing depolarization or hyperpolarization of neurons to activate or inhibit them. Magnetogenetics manipulates neuronal activity through electromagnetic nanoparticles and the opening and closing of heat-sensitive channels like transient receptor potential vanilloid 1 (TRPV1). Chemogenetic technology prominently features designer receptors exclusively activated by designer drugs (DREADDs), where human muscarinic acetylcholine receptor subtype M3 (hM3Dq) excites neurons under clozapine-N-oxide (CNO) stimulation, while human muscarinic acetylcholine receptor subtype M4 (hM4Di) induces neuronal inhibition.

**Figure 2 cells-14-00122-f002:**
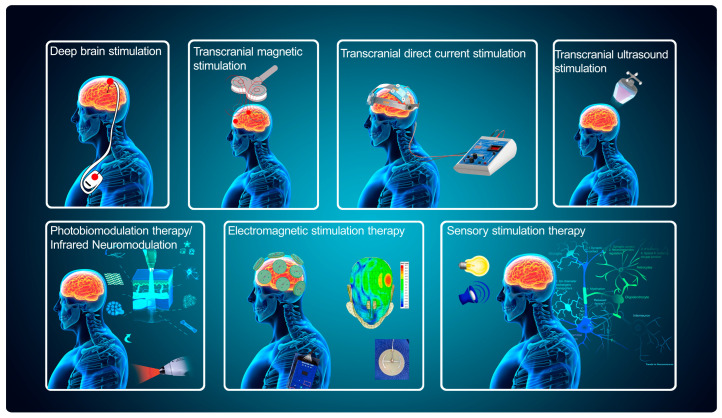
Schematic diagram of non-genetic physical factor neuromodulation technologies. Non-genetic physical factor neuromodulation technology includes deep brain stimulation (DBS), transcranial magnetic stimulation (TMS), transcranial direct current stimulation (tDCS), transcranial ultrasound stimulation (TUS), photobiomodulation therapy (PBMT), infrared neuromodulation (INM), electromagnetic stimulation therapy, and sensory stimulation therapy. DBS involves placing bipolar electrodes in specific brain regions and using implantable pulse generators to stimulate certain neural nuclei or regions deep in the brain, correcting abnormal neural circuit activities and alleviating neuropsychiatric symptoms. TMS induces local currents in the cerebral cortex through strong pulsed magnetic fields to regulate neuron electrical activity and electrical signal transmission. tDCS involves placing two or more electrodes on the surface of the scalp and applying a weak current to promote the interaction of neurotransmitters in the brain and induce changes in synaptic plasticity. PBMT (also known as low-intensity laser therapy) is based on the mechanism of photobiomodulation and includes both visible and near-infrared light neuromodulation techniques (figure revised from [129]). Electromagnetic stimulation therapy projects electromagnetic waves into the brain through a transmitter, thereby modulating cortical and subcortical neural activity (figure revised from [130]). Sensory stimulation therapy employs precisely calibrated sensory inputs, such as sound waves and visible light, to induce and modulate rhythmic neuronal activity in the brain’s primary sensory areas. This rhythmic activity, characterized by synchronized oscillations of neural populations at specific frequencies (particularly in the gamma range of 30–100 Hz), plays a crucial role in neural communication, sensory processing, and cognitive function.

**Table 1 cells-14-00122-t001:** Comparison of advantages and disadvantages of existing neuromodulation technologies.

Neuromodulation Technology		Advantage	Disadvantage
Genetic neuromodulation technology	Optogenetics	High time precision, the speed of regulating target cells can reach the millisecond level.	Associated physical restraints alter natural behavior; The control part is limited to the position close to the optical fiber; The fiber-optic implant strategy is more traumatic.
	Chemogenetics	Controllability and flexibility; Simple operation; Less trauma.	Low time precision; The precision of the stimulus intensity is not high.
	Sonogenetics _	Safe and noninvasive; Small focus area; No implants are needed.	Head mounting fixtures; Mechanical effects and cavitation effects.
	Magnetogenetics	Good targeting; Less traumatic.	The mechanism has been unclear; Resonant coil near the head; It is difficult to develop an alternating magnetic field generator suitable for the human body; Poor timeliness.
Non-genetic physical neuromodulation technology	Deep Brain Stimulation	Reversibility; Adjustability.	Invasive; permanent damage to brain tissue and overlying skull/scalp due to implantation of electrodes; The executors are required to have a high level of surgical operation; Parameters need to be adjusted for each stimulation to match disease progression characteristics; Chronic immune response induced at the implant–tissue interface.
	Transcranial Magnetic Stimulation	Noninvasive adjustment; Patients are able to be treated at home.	Difficulty targeting specific cell types or brain regions; The magnetic field has a thermal effect on biological tissue.
	Transcranial Direct Current Stimulation	Painless; Noninvasive; Easy to operate; Safe and reliable.	Difficulty targeting specific cell types or brain regions with coarse spatial precision; Electrical stimulation brings a certain thermal effect on biological tissues.
	Transcranial Ultrasound Stimulation	High spatial precision to target shallow or deep regions.	Low temporal accuracy; Expensive; Needs to be implemented in a hospital; Efficacy remains uncertain.
	Photobiomodulation Therapy	Affordable cost; Suitable for home-based application.	Limited spatial and temporal precision; Confined to superficial cortical stimulation; Efficacy remains uncertain.
	Electromagnetic Therapy	Penetrates deep brain regions; Suitable for home-based application.	Limited temporal precision; Uncertain therapeutic efficacy.
	Gamma ENtrainment Using Sensory stimulation	Affordable cost; Enables modulation of brain activity in relevant regions through endogenous processes.	Confined to specific brain regions.

**Table 2 cells-14-00122-t002:** Studies related to the regulation of cell activity by acoustic genetics.

	Dosage Information	Target Site	Medium	Effect	Reference
Modulation of mechanosensitive ion channels by sound waves to regulate cell activity	10 ms single pulse, 2.25 MHz low-pressure (peak negative pressure level below 0.5 MPa) ultrasound	Mechanosensitive neurons expressing the *Caenorhabditis elegans* gene (*TRP-4*)	*TRP-4*	Induction of bending and reversing behavior in *Caenorhabditis elegans*.	Ibsen/2015 [60]
	Pulsed ultrasound with repetition rate in the range from 30 Hz to 10 kHz	*Caenorhabditis elegans* carrying *TRP-4*, MEC-3, MEC-4 mutants	MEC-3 MEC-4	Focused ultrasound elicits reversal behavior in freely moving *TRP-4* (but not MEC-3, MEC-4) mutant-carrying nematodes in a pressure- and stimulus-duration-dependent manner.	Kubanek/2018 [62]
	7 MHz, 2.5 MPa, 100 ms duration	Neurons expressing exogenous hs *transient receptor potential ankyrin 1 (TRPA1)* in the left primary motor cortex	*TRPA1*	The right extremity showed ultrasound-dose-dependent EMG responses and movements.	Duque/2022 [63]
	Low-pressure (0.25 MPa) pulsed ultrasound	Primary hippocampal neurons expressing mechanosensitive channel of large conductance (MSCL)	MscL	Timed short pulses can elicit single spikes of activity in neurons.	Ye/2018 [64]
	3.2 MHz fundamental frequency, 1.6 MPa peak negative pressure, 50% duty cycle, 10 Hz repetition rate, 10 stimulation time, 10 s interval ultrasound	*Transient receptor potential melastatin 2 (TRPM2)* highly expressed neurons in the preoptic area of the hypothalamus	*TRPM2*	The skin temperature of the mouse scapula decreased significantly, and the temperature of the tail increased significantly, entering a reversible dormancy-like state.	Yang/2023 [65]
	0.5 MHz center frequency, 400 to 500 μs pulse width, 300 ms stimulation duration, 0.05 to 0.2 MPa pressure range, 1 kHz repetition rate pulsed ultrasound	*MscL* ion channel in the subthalamic nucleus	MscL	Significantly alleviated the motor symptoms of PD model mice and improved the motor coordination of mice.	Xian/2023 [66]
Regulation of locally responsive cellular activity by means of ultrasound-sensitive proteins/nanoparticles	Low-intensity focused ultrasound	Mouse-specific neurons that selectively express *transient receptor potential vanilloid 1 (TRPV1)* in the striatum	*TRPV1*	Repeatedly evoked rotational behavior in freely moving mice.	Yang/2021 [61]
	3.75 MHz frequency, 7.6 V peak continuous wave ultrasonic	Human Embryonic Kidney (HEK) 293T cells transfected with ultrasonic actuator balloons	Ultrasonic Actuator Airbag	Selective acoustic sorting of mammalian cells.	Wu/2023 [67]

**Table 3 cells-14-00122-t003:** Related research on magnetic regulation of cell activity.

	Dosage Information	Target Site	Medium	Effect	Reference
Endogenous nanoparticles for gene editing	465 kHz (5 mT)	HEK cells, pheochromocytoma (PC12) cells, tumor-bearing mice	FeNPs^anti-His^-TRPV1^His^, Ferritin-TRPV1	Influx of calcium ions, increased insulin release	Stanley/2012 [83]
	465 kHz (32 or 29 mT) or static magnetic field with magnetic force >10 pN	HEK/MSCs/mouse	Anti-green fluorescent protein (GFP)-TRPV1/GFP- ferritin	Regulates insulin release, regulates glucose homeostasis	Stanley/2015 [84]
	465 kHz (31, 27 and 23 mT) static magnetic field (cell: 280 mT, 130 mT) (mouse: 0.5–1 T or 0.2–0.5 T)	Neuroblastoma (N38) cells/Cre mouse	Anti-GFP-TRPV1/GFP-ferritin	Bidirectional regulation of feeding and metabolism by the hypothalamus	Stanley/2016 [85]
	0–27 mT (Requires a power supply of 5 V and 400 mA)	HEK cells	Anti-GFP-TRPV1/GFP-ferritin	Increased calcium ion influx; in Rho-positive cells, it can increase the number of cell protrusions, promote wound healing, and control the direction of cell growth	Mosabbir/2018 [86]
	Brain slices (50 mT static magnetic field) Zebrafish (500 mT) Rat (50–250 mT)	HEK/brain slice entorhinal cortex/zebrafish sensory neurons/rat striatum dopaminergic neurons	Ferritin-TRPV4 fusion protein (Magneto 2.0)	Calcium ion influx/increased firing rate/increased calcium ions in zebrafish neurons, behavioral changes/reward-related behavioral changes in rats	Wheeler/2016 [87]
	200 mT	HEK cells	Magneto 2.0	Ca^2+^ influx	Dure/2019 [88]
	Four 50 ms square wave pulses at 20 V each (175 MHz, 36 µT)	Chick/zebrafish neural crest cells	Inserting a ferritin-binding motif into *TRPV1/4* channels	Craniofacial/cardiac birth defects due to maternal fever	Hutson/2017 [89]
Magnetothermal effect of nanoparticle on cell surface	40 MHz, 8.4 G	(*TRPV1*-expressing) HEK 293T cells/rat primary hippocampal neurons/*Caenorhabditis elegans*	MnFe_2_O_4_ -TRPV1	Calcium influx/escape behavior in *Caenorhabditis elegans*	Huang/2010 [90]
	450 W/g at 500 kHz and 15 kA/m (18.75 mT)	HEK cells, neurons, mice	Superparamagnetic cobalt manganese ferrite nanoparticles–TRPV1	Magnetothermal stimulation of the motor cortex induces walking, stimulation of the striatum induces body rotation, and stimulation near the ridge between the ventral and dorsal striatum induces freezing of gait	Munshi/2017 [91]
	ƒ = 500 kHz and field amplitude Ho = 15 kA/m (18.75 mT)	HEK cells, primary hippocampal neurons, mice	Superparamagnetic Nanomaterial–TRPV1	Ventral tegmental area neurons are awakened by activation of TRPV1-positive cells	Chen/2015 [92]
	80 mT, 49.9 kHz alternating magnetic field; 12 mT, 555 kHz alternating magnetic field	(expressing *TRPA1*-A)	15 nm codoped iron oxide nanoparticles with high coercive force; undoped 40 nm iron oxide nanoclusters with low coercive force	Rapid control of wing posture in Drosophila	Sebesta/2022 [93]
Mechanical force	Force (pN)	Cell mechanosensitive receptors Notch and E-cadherin	Zn-doped ferrite, core and plasmonic gold shell functionalized with benzylguanine	Activation of cell surface mechanoreceptors	Seo/2016 [94]
	Static magnetic field/5 Hz square wave current/pulse frequency 100–1000 Hz	Inner-ear hair cells	Cubic magnetic nanoparticles (bound to membrane endogenous glycoproteins)	Stereocilia are noncontact mechanically controlled, generating displacements of tens of nanometers that result in ion influx into hair cells	Lee/2014 [95]
	Force (pN)	Embryonic chicken forebrain neurons	Immunomagnetic beads coated with chicken b1 integrin	Activates neurons, altering neurite outgrowth; speed of force application determines facilitation (slow) or inhibition (fast) of neurite	Fass/2003 [96]
	190–270 pN 300 pN (magnetic field gradient 10,000 T/m)	Cortical neurons	Highly parallelized nanomagnetic materials on a chip	Intracellular Tau protein redistribution and cellular displacement	Kunze/2015 [97]
	15 pN	Retinal ganglion cells	Superparamagnetic nanoparticles coated with Trkb antibody	Affects axon growth	Steketee/2011 [98]
	Force (pN) Nanomagnetic (0.1–1 nN) (Maximum magnetic field strength 150 mT)	Cortical neuron	Ferromagnetic nanoparticles coated with starch or chitosan	Ca influx	Tay/2016 [99]
	1588 mM^−1^S^−1^ (in 1 wt% agar phantom at a field strength of 7 T at 300 K)	Liver tumor site	Immunomodulatory magnetic microspheres synthesized from cubic iron oxide nanoparticles, recombinant interferon-γ, and biodegradable polylactide-co-glycolide (PLGA)	Induction of natural killer (NK) cell infiltration into liver tumor sites	Park/2017 [100]
	Force An external magnetic field of approximately 0.15 T applied by two permanent NdFeB magnets	Human umbilical vein endothelial cells (HUVEC)	Ferrite magnetic nanoparticles doped with Zn^2+^	Induces intracellular signaling processes, leading to angiogenesis	Goya/2014 [101]
	Force (n N) over 100 nN and 5 nN μm^−1^	HeLa cell cortex	Magnetic nanoparticles	Activation of the mechanotransducer p21-activated kinase (PAK), induces asymmetry in filopodia and disrupts adjacent actin stress fibers	Tseng/2012 [102]
Aggregate-activated cell signaling	At least 0.3 T	Colon cancer cells	Zinc-doped iron oxide magnetic nanoparticles (Zn_0.4_Fe_2.6_O_4_)	Binds to an antibody targeting DLD-1 colon cancer cell death receptor 4 and promotes the apoptotic signaling pathway	Cho/2012 [103]
	Five electromagnetic pulses of 1-min duration (0.1 A, 0.3 A and 1 A) separated by 1-min rest periods—	Mast cells	DNP-lysine-labeled superparamagnetic nanoparticles	Intracellular calcium influx	Mannix/2008 [104]
	Static magnetic field	T cells	Magnetic nano iron dextran (NanoaAPC)	Increases the accumulation of T-cell receptors and slows the rate of tumor growth	Perica/2014 [105]

## Data Availability

Not applicable.

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
