# Peer review of "Advancing Neuroscience and Therapy: Insights into Genetic and Non-Genetic Neuromodulation Approaches"

_cells, 2025, doi:10.3390/cells14020122_

Round 1
Reviewer 1 Report (Previous Reviewer 1)
Comments and Suggestions for Authors
The authors addressed all my questions and the review is much better now.
Just a very minor aspect. I would suggest to replace "pain management" by "pain relief".
Author Response
Dear Editors and Reviewers:
Thank you for your letter and the reviewers’ comments concerning our manuscript entitled “Advancing Neuroscience and Therapy: Insights into Genetic and Non-genetic Neuromodulation Approaches” (ID: cells-3395554). Those comments are valuable and helpful for revising and improving our paper and the important guiding significance to our research.
In the following, we present a detailed response to all the comments raised by the reviewer point-by-point, together with the relevant revisions. All changes are also highlighted in red in the manuscript. We thank the editor and experts again for considering our work and believe the modifications have made the manuscript clearer and more accurate. If there are any other modifications we could make, we would like very much to modify them, and we really appreciate your help.
Best wishes,
Yours sincerely,
Weijia Zhi, Ying Li, Lifeng Wang, Xiangjun Hu
Laboratory Experimental Pathology
Beijing Institute of Radiation Medicine
Suggestions From Reviewer 1|Round 1
Reviewer 1’s Comment 1:
The authors addressed all my questions and the review is much better now.
Just a very minor aspect. I would suggest to replace “pain management” by “pain relief”.
Response to comment:
Thank you very much for your valuable suggestion. We have replaced the term “pain management” with “pain relief” as per your recommendation. We believe this change enhances the clarity and accuracy of the text. We appreciate your thoughtful input.
The revisited sentences are as follows:
This non-invasive technique is versatile, modulating neuronal activity in both research and clinical settings, such as auditory restoration in cochlear implants [197], pain relief [198], muscle control [198], and treatment of neurological conditions like epilepsy and PD [197,199,200].
Location: Page 30, Line 1313.

Reviewer 2 Report (Previous Reviewer 3)
Comments and Suggestions for Authors
The authors have provided clarification of some terms including acronyms that might be unfamiliar to non-experts. Overall, the clarifications have improved the manuscript substantially. Nonetheless some acronyms remain undefined. For example,
Line 318 The acronym GPCR is not defined. (It was defined in previous version, but the definition was within text deleted from the current version.)
Some sentences are long and difficult to understand, Foe example: Lines 1588- 1592 contain a long sentence with clumsy grammar that might be easier to understand if abbreviated
Line 1598 ‘ However, the clinical translation of existing neuromodulation technologies remains confronted with numerous challenges’ could be simplified to ‘However, the clinical translation of existing neuromodulation technologies presents numerous challenges’
Comments on the Quality of English LanguageSome sentences are long and difficult to understand, Foe example: Lines 1588- 1592 contain a long sentence with clumsy grammar that might be easier to understand if abbreviated
Line 1598 ‘ However, the clinical translation of existing neuromodulation technologies remains confronted with numerous challenges’ could be simplified to ‘However, the clinical translation of existing neuromodulation technologies presents numerous challenges’
Author Response
General Response
Dear Editors and Reviewers:
Thank you for your letter and the reviewers’ comments concerning our manuscript entitled “Advancing Neuroscience and Therapy: Insights into Genetic and Non-genetic Neuromodulation Approaches” (ID: cells-3395554). Those comments are valuable and helpful for revising and improving our paper and the important guiding significance to our research.
In the following, we present a detailed response to all the comments raised by the reviewer point-by-point, together with the relevant revisions. All changes are also highlighted in red in the manuscript. We thank the editor and experts again for considering our work and believe the modifications have made the manuscript clearer and more accurate. If there are any other modifications we could make, we would like very much to modify them, and we really appreciate your help.
Best wishes,
Yours sincerely,
Weijia Zhi, Ying Li, Lifeng Wang, Xiangjun Hu
Laboratory Experimental Pathology
Beijing Institute of Radiation Medicine
Suggestions From Reviewer 2|Round 1
Reviewer 2’s Comment 1:
The authors have provided clarification of some terms including acronyms that might be unfamiliar to non-experts. Overall, the clarifications have improved the manuscript substantially. Nonetheless some acronyms remain undefined. For example, Line 318 The acronym GPCR is not defined. (It was defined in previous version, but the definition was within text deleted from the current version.)
Response to comment:
We sincerely thank the reviewer for their thoughtful feedback. In response to your comment regarding the acronym GPCR, we have carefully reviewed the manuscript and added the full form of G-protein coupled receptor (GPCR) on Line 314. Furthermore, we have conducted a comprehensive check of all acronyms throughout the manuscript, ensuring that each is properly defined at its first occurrence. To further assist readers, we have compiled a list of all abbreviations and their full forms in a table, which has been placed at the end of the manuscript (see Table 4 below or in the manuscript). We hope these revisions improve the accessibility and clarity of the manuscript for non-expert readers, as per your suggestion.
Reviewer 2’s Comment 2:
Some sentences are long and difficult to understand, Foe example: Lines 1588-1592 contain a long sentence with clumsy grammar that might be easier to understand if abbreviated
Line 1598 ‘However, the clinical translation of existing neuromodulation technologies remains confronted with numerous challenges’ could be simplified to ‘However, the clinical translation of existing neuromodulation technologies presents numerous challenges’
Response to comment:
We greatly appreciate the reviewer’s careful reading and valuable suggestions regarding sentence structure. As per your recommendation, we have revised the sentence on Line 1595 to: "However, the clinical translation of existing neuromodulation technologies presents numerous challenges." In addition to this, we have conducted a thorough review of the entire manuscript, identifying and addressing other instances of long or convoluted sentences. All such sentences have been revised for greater clarity, ensuring that the language is more concise and accessible. We believe these revisions significantly enhance the readability and overall quality of the manuscript.
Moreover, all changes are highlighted in red in the manuscript.
Table 4. List of Abbreviations and Corresponding Full Terms.
|
Abbreviation |
Full Term |
|
AAV |
Adeno-Associated Virus |
|
AC |
Auditory Cortex |
|
AD |
Alzheimer’s Disease |
|
Arch |
Archaeal Halorhodopsin |
|
α7nACHR |
Alpha 7 nicotinic Acetylcholine Receptor |
|
BDNF |
Brain-Derived Neurotrophic Factor |
|
BOLD |
Blood Oxygen Level-Dependent |
|
GECI |
Genetically Encoded Ca2+ Indicators |
|
ChR2 |
Channelrhodopsin-2 |
|
CINP |
International College of Neuropsychopharmacology |
|
CNO |
Clozapine-N-oxide |
|
CNS |
Central Nervous System |
|
CREB |
cAMP Response Element-binding Protein |
|
CRISPR |
Clustered Regularly Interspaced Short Palindromic Repeats |
|
DAAO |
D-amino Acid Oxidase |
|
dACC |
Dorsal Anterior Cingulate Cortex |
|
DBS |
Deep Brain Stimulation |
|
dlPFC |
Dorsolateral Prefrontal Cortex |
|
DMT1 |
Divalent Metal-ion Transporter-1 |
|
DREADDs |
Designer Receptors Exclusively Activated by Designer Drugs |
|
dTMS |
Deep Repetitive Transcranial Magnetic Stimulation |
|
eBR |
Engineered Bacteriorhodopsin |
|
ECM |
Extracellular Matrix |
|
EEG |
Electroencephalography |
|
EMG |
Electromyography |
|
EMT |
Electromagnetic Therapy |
|
FDA |
Food and Drug Administration |
|
fMRI |
Functional Magnetic Resonance Imaging |
|
GABA |
Gamma-Aminobutyric Acid |
|
GAD |
Generalized Anxiety Disorder |
|
GENUS |
Gamma ENtrainment Using Sensory stimulation |
|
GFP |
Green Fluorescent Protein |
|
Gipr |
Glucose-Dependent Insulinotropic Polypeptide Receptor |
|
GPCR |
G-Protein-Coupled Receptor |
|
GVs |
Gas Vesicles |
|
HD |
Huntington’s Disease |
|
HEK |
Human Embryonic Kidney |
|
HIF-1α |
Hypoxia-Inducible Factor 1-alpha |
|
hM3Dq |
human muscarinic acetylcholine receptor subtype M3 |
|
hM2Di |
human muscarinic acetylcholine receptor subtype M2 |
|
hM4Di |
human muscarinic acetylcholine receptor subtype M4 |
|
IL |
Interleukin |
|
INI |
Infrared Neural Inhibition |
|
INM |
Infrared Neuromodulation |
|
INS |
Infrared Neural Stimulation |
|
IR |
Infrared |
|
K2P |
Two-Pore Domain Potassium Channels |
|
LIFUS |
Low-Intensity Focused Ultrasound |
|
LTD |
Long-Term Depression |
|
LTP |
Long-Term Potentiation |
|
mPFC |
Medial Prefrontal Cortex |
|
MRI |
Magnetic Resonance Imaging |
|
MSC |
Mechanosensitive Ion Channels |
|
MSCL |
Mechanosensitive Channel of Large Conductance |
|
NAcc |
Nucleus Accumbens |
|
NBT |
Navigated Brain Therapy |
|
NIR-â…¡ |
Second Near-Infrared Spectral Window |
|
NMDA |
N-Methyl-D-Aspartate |
|
NpHR3.0 |
Natronomonas Halorhodopsin 3.0 |
|
OCD |
Obsessive-Compulsive Disorder |
|
OFC |
Orbital Frontal Cortex |
|
OptoXR |
Optogenetic G protein-Coupled Receptor |
|
OxPhos |
Oxidative Phosphorylation |
|
PBMT |
Photobiomodulation Therapy |
|
PD |
Parkinson’s Disease |
|
PEG |
Polyethylene Glycol |
|
PET |
Positron Emission Tomography |
|
pm-AuNRs |
Plasma Membrane-targeted Gold Nanorods |
|
PSAM |
Pharmacologically Selective Actuator Module |
|
PSEM |
Pharmacologically Selective Effector Molecule |
|
POA |
Preoptic Area |
|
RASSLs |
Receptors Activated Solely by Synthetic Ligands |
|
RF |
Radio Frequency |
|
RNS |
Responsive Neurostimulation |
|
ROS |
Reactive Oxygen Species |
|
rTMS |
Repetitive Transcranial Magnetic Stimulation |
|
S1 |
Somatosensory Cortex |
|
SCG |
Subcallosal Cingulate Gyrus |
|
tDCS |
Transcranial Direct Current Stimulation |
|
TEMT |
Transcranial Electromagnetic Therapy |
|
TI |
Temporal Interference |
|
TMAS |
Transcranial Magnetoacoustical Stimulation |
|
TMS |
Transcranial Magnetic Stimulation |
|
TRD |
Treatment-resistant Depression |
|
TREK |
TWIK-related K+ Channel |
|
TRP |
Transient Receptor Potential |
|
TRPA1 |
Transient Receptor Potential Ankyrin 1 |
|
TRPC |
Transient Receptor Potential Canonical |
|
TRPM2 |
Transient Receptor Potential Melastatin 2 |
|
TRPML |
Transient Receptor Potential Mucolipin |
|
TRPN |
Transient Receptor Potential Drosophila NOMPC |
|
TRPP |
Transient Receptor Potential Polycystin |
|
TRPV1 |
Transient Receptor Potential Vallinoid 1 |
|
TRPV4 |
Transient Receptor Potential Vanilloid 4 |
|
TUS |
Transcranial Ultrasound Stimulation |
|
VChR1 |
Volvox Channelrhodopsin-1 |
|
VC/VS |
Ventral Internal Capsule and Ventral Striatum |
|
VGCs |
Voltage-gated Channels |
|
VIP |
Vasoactive Intestinal Peptide |
|
WHO |
World Health Organization |
|
Y-BOCS |
Yale-Brown Obsessive Compulsive Scale |
|
Zn0.4Fe2.6O4 |
Zinc-doped Iron Oxide Magnetic Nanoparticles |

This manuscript is a resubmission of an earlier submission. The following is a list of the peer review reports and author responses from that submission.
Round 1
Reviewer 1 Report
Comments and Suggestions for Authors
This is a very nice and detailed review that describes different neuromodulation strategies to treat neurological disorders. The review is clear and well written, and I have just minor considerations:
1. The authors often use the term ‘psychiatric disorders’, but I think it is better to use ‘neurological disorders’ since it is more general and includes diseases that have a clear genetic origin or neuronal malfunction, such as Huntington’s or some forms of Parkinson’s. In the abstract, line 19, I think ‘neurological disorders’ is more appropriate.
2. Figure 1 is the only figure in the review. Fonts appear very small when printed out. I would suggest splitting the figure in two, so each panel appears in the corresponding section. Also make the fonts bigger for comfortable reading.
3. I did not find important grammar or spelling errors, but I think it is better to avoid the use of contractions, such as “it’s” on lines 103 and 483.
4. On line 82, I think ‘strategies’ is a better word than ‘tactics’.
5. The call to Tables 1 to 3 have to be revised. Table 3 is called first, then Table 1 and 2. Please be sure that in Latex they are organized and called in an ascending manner.
6. Authors mention therapies related to Parkinson’s, but nothing is said about Huntington’s, which is a disease that has also strong prominence. The authors could cite a recent paper on neuromodulation in Huntington (such as Fernández-García et al., “M2 cortex-dorsolateral striatum stimulation reverses motor symptoms and synaptic deficits in Huntington’s disease”, eLife 2020; or others) and/or reviews in the field (such as L. Jose et al., “Non-invasive neuromodulation methods to alleviate symptoms of Huntington's disease: a systematic review of the literature”, Journal of Clinical Medicine 2023; or others).
7. The authors could include neuromodulation or infrared neuromodulation as an additional technique, i.e., the control of neuronal activity with heat. It is not as famous as optogenetics or other techniques but it is gaining importance.
8. At the beginning of Section 2.4.2, I think a small paragraph is need to mention that two main regulatory mechanisms will be considered. Otherwise, the sudden titles that appear next look like out of context.
9. The title of Section 3 should say “technologies” instead of “technology”.
10. In line 1287, I think “implant substance” can be replaced by “implanted device” a more appropriate word than “substance”.
11. In line 1336, “mechanism” should be plural.
Comments on the Quality of English Language
The use of English language is correct. The manuscript reads very well and I detected only very minor errors.
Author Response
Please see the attachment. Thank you, and have a good day!

Reviewer 2 Report
Comments and Suggestions for Authors
The purpose of this review is to provide an overview and evaluation of different neuromodulation techniques, including sonogenetics, magnetogenetics, optogenetics, and chemogenetics. The review aims to discuss the principles, mechanisms, advantages, limitations, and potential applications of these techniques in studying neural circuits, understanding brain function, and developing therapeutic interventions for neurological disorders. It also highlights the current challenges and areas for further research and development in order to optimize the effectiveness, specificity, and safety of these techniques. The review also touches on the ethical considerations associated with these techniques. Paragraphing is concise and good, and the article consists of major recent advancements in the field of neuromodulation and deserve publication after some revisions
My main suggestions to the Authors is as follows:
1. Abstract: I suggest the authors to rewrite the abstract consisting of the major aspects of the entire paper in a prescribed sequence that includes: 1) the overall purpose of the study and the research problems you investigated; 2) the basic design of the study; 3) major findings in sequence.
2. Previously, Neuromodulation therapies, including transcranial magnetic stimulation (TMS), transcranial electrical stimulation (TES), electroconvulsive therapy (ECT), photobiomodulation (PBM), transcranial ultrasound stimulation (TUS), deep brain stimulation (DBS), and vagus nerve stimulation (VNS), all have been reported to attenuate neuroinflammation and reduce the release of pro-inflammatory factors. However, the authors did not discuss about the beneficial effects of neuromodulation in the review. I suggest the authors to briefly discuss about this.
3. The questions that author missed to address in conclusion:
There is always a dilemma on how to conclude a review article. Since the authors have deliberately summarized huge amounts of published results, it will go a long way. It would be helpful if they can provide their own thoughts that would in turn help in finding the areas that need to be addressed. For example, what are the factors that one needs to consider and what research is necessary to make a clinical translation of neuromodulation techniques more successful. What limitations are hindering their clinical translation and in what direction does the future research need to be, to make the clinical translation possible?
Author Response

(The authors gave the same response as above.)

Reviewer 3 Report
Comments and Suggestions for Authors
The authors provide an over-ambitious comprehensive review of genetic and non-genetic neuromodulation techniques. They include a great deal of detail about many different techniques, but they fail to provide concise description of most of the genetic techniques. It would be helpful if a concise statement of the relevant techniques was provided at the beginning of each of the sections on genetic neuromodulation. There are many dense sentences packed with technical terms that are not defined. For example:
Line 371 There is no definition of ‘up conversion of micro-nano particles’
Line 456 the term ‘conduction force of the cytoskeleton’ is not defined
Sonogenetic and non-genetic sonic stimulation techniques are mingled in a single section without clear distinction
Nanoparticles are not explicitly defined until line 552, despite numerous references to these particles in previous sections
Section 2.4 (magnetogenetics) As far as I can understand, the techniques described in 2.4.2 involve nanoparticle implantation, but it is not clear which of the studies described involve genetic modification. For example in Line 693 that authors describe using nanoparticles to target cells expressing the temperature-sensitive ion channel TRPV1. It is not clear whether they are describing a technique in which the target cells have been genetically modified so that they express TRPV1 or alternative, that the target cells express endogenous TRPV1 . Finally near the end of a long section on magnetogenetics, the authors provide a summary on lines 806-809 that implies that the techniques involve modification of the TPRV1 receptor.
With regard to the genetic techniques, they fail to provide clear indication of which techniques are applicable in humans. In cases where it appears they are describing human applications they do not provide information about the procedure for modifying genes in humans.
in the section on non-genetic techniques the authors include too much material in an oversimplistic manner. In places this leads to very misleading statements. For example, they state that tDCS is often used to treat Alzheimer’s disease (AD). The authors justify this statement with a reference to a review of the use of tDCS to treat Alzheimer’s disease by Madji et al. In fact Madji at al conclude : ‘However, due to the small number of studies and the high heterogeneity of the data, more high-quality studies using standardized parameters and measures are needed before tDCS can be considered as a treatment for AD.
On numerous occasions acronymns are used without definition. Foe example:
P11, line 351 TRPs are not defined
Line 464 LIFUS is not defined.
Line 1084: TUS is introduced without definition
Line 1296 AAV is introduced without definition
Overall the article would be far more informative if the authors did not attempt to cram in so many obscure details regarding genetic techniques but instead provided a clear description of the general principles.
Comments on the Quality of English Language
In many instances use of language is imprecise. For example:
P3 line 108 ‘’Optogenetics entails the introduction of light-sensitive genes …. into precise nervous system cells’’ I presume they mean ’precisely identified nervous system cells’
P4 132 ‘Optogenetics can regulate target cells at a speed of milliseconds’ I presume they mean ‘on a time scale of milliseconds’ ;
Line 377 ‘Ultrasound can bind to genes’ The meaning is unclear
Line 919 ‘the implementation of DBS requires the performer to have a high level of surgical operation’; I presume they mean expertise
Author Response

(The authors gave the same response as above.)

Round 2
Reviewer 2 Report
Comments and Suggestions for Authors
I do not have any further concerns related to this manuscript and authors have addressed all the questions that I have raised.
Reviewer 3 Report
Comments and Suggestions for Authors
As noted in my review of the earlier version of this manuscript, the authors provide an over-ambitious comprehensive review of genetic and non-genetic neuromodulation techniques. In the revised version the authors have improved the manuscript by providing definitions of some terms and descriptions of techniques, as suggested in my previous review. In that review I had stated that there were widespread instances of unclear language, but I did not attempt the daunting task of identifying all the numerous instances in which the text lacked clarity. I consider that the authors should take responsibility for addressing this daunting task. In this review I will again provide some explicit instances of lack of clarity, but it would be a major task to re-write the text in a clear manner.
In the revision the acronyms MSCs and HEK are used on multiple occasions, but are not explicitly defined
In many instances the use of words is likely to create confusion in the mind of an intelligent non-expert reader. For example, in line 968, the significance of the term ‘genetically encoded’ in the phrase ‘genetically encoded endogenous metal nanoparticles’ is unclear. In what sense are the endogenous nanoparticles genetically encoded? All naturally occurring (i.e. ‘endogenous’) proteins in the body are coded in genes. In the appropriate milieu, these proteins naturally fold in a manner that a creates site with the shape and electrical properties required to bind relevant metal atoms or ions. However, it is not clear which of the techniques described in this manuscript involve a manipulation that modulates the expression of the relevant proteins.
In line 942 the phrase ‘applying a weak current to promote the interaction of neurotransmitters’ is vague. What is meant by ‘interaction of neurotransmitters’ ? Do they mean interaction between neurons expressing different neurotransmitters?
As suggested in my previous review, the article would be far more informative if the authors did not attempt to cram in so many obscure details regarding genetic techniques but instead provided a clear description of the general principles. Perhaps it would be helpful if the authors engaged a collaborator with competence in writing clearly for an intelligent but non-expert readership.
Comments on the Quality of English LanguageAs noted in my comments for the authors, the text us unclear. This is largely due to inadequate defintion of the technical terms rather than poor quality of English language, though in places, better use of English might facilitate clarity.